# The place-cell representation of volumetric space in rats

Roddy M. Grieves [1]*, Selim Jedidi-Ayoub[1], Karyna Mishchanchuk[1], Anyi Liu[1], Sophie Renaudineau[1] & Kate J. Jeffery[1]*

Place cells are spatially modulated neurons found in the hippocampus that underlie spatial memory and navigation: how these neurons represent 3D space is crucial for a full understanding of spatial cognition. We wirelessly recorded place cells in rats as they explored a cubic lattice climbing frame which could be aligned or tilted with respect to gravity. Place cells represented the entire volume of the mazes: their activity tended to be aligned with the maze axes, and when it was more difficult for the animals to move vertically the cells represented space less accurately and less stably. These results demonstrate that even surface-dwelling animals represent 3D space and suggests there is a fundamental relationship between environment structure, gravity, movement and spatial memory.

---

[1] University College London, Institute of Behavioural Neuroscience, Department of Experimental Psychology, London, UK. *email: rmgrieves@gmail.com; k.jeffery@ucl.ac.uk

Place cells are neurons in the hippocampus that fire when an animal visits specific regions of its environment, called place fields, and are thought to provide the foundation for an internal representation of space, or 'cognitive map'[1,2]. The question arises as to whether this map is three-dimensional, and if so whether its properties are the same in all dimensions, and how information is integrated across these dimensions[3–5]. This is important not just for spatial mapping per se but also because the spatial map may form the framework for other types of cognition in which information dimensionality is higher than in real space. Understanding how the brain integrates information across dimensions is thus of theoretical importance.

A previous study of place cells in rats[6] found vertical elongation of the place fields when rats climbed either a pegboard wall studded with footholds or a helical track, suggesting that perhaps the cognitive map has a lower resolution for vertical space than for horizontal space (i.e., is anisotropic). This finding was supported by observations that entorhinal grid cells, thought to provide a spatial metric for place cells, showed absent spatial processing in the vertical dimension. However, in a more recent experiment, when rats climbed a wall covered with chicken wire, which oriented them parallel to the wall instead of the floor, place cells were found to have normally shaped firing fields, although fields themselves occurred with lower probability than on the floor[7]. This meant that although the firing of spatial neurons differed between the floor and the wall, the horizontal and vertical components of firing on the wall did not appreciably differ. Taking these findings together, it seems that the differences in spatial encoding previously seen in the vertical dimension may be due to the different constraints on movement, or the locomotor 'affordances' in the different dimensions[8]. Meanwhile, studies in flying bats have reported 3D place fields[9] that do not deviate statistically from spherical[10], suggesting a spatial map of equal resolution in all dimensions (isotropic).

The different patterns of neural activity in the different types of apparatus could be due to the different movement patterns afforded by the footholds (aligned vs. orthogonal to gravity), or to the different encoding requirements of traveling on a surface vs. through a volume. The present experiment aimed to untangle these issues by exploring, in rats, the interaction between gravity, which is what distinguishes horizontal from vertical, and the locomotor affordances of the environment. Animals were recorded using digital telemetry as they explored a volumetric space—an open cubic lattice—through which they could move freely and which had equal properties in all three spatial dimensions. Place cells exhibited firing fields throughout this volume, confirming that these cells underlie a fully three-dimensional volumetric representation of space. Furthermore, we found that place fields tended to be elongated along the axes of the maze (the directions aligned with the boundaries, and in which travel was easiest) with greater elongation for the vertical axis and a resultant lower spatial information and decoding accuracy. We then tilted the lattice so that the three planes of movement all had the same relationship to gravity, and were thus all equally easy (or hard) to traverse. We found that the elongation of the axes followed the tilt of the maze, and the difference between horizontal and vertical place field metrics disappeared. Thus, differences between horizontal and vertical metrics in spatial encoding seem to arise from the greater movement constraints for vertical than horizontal travel, rather than from an intrinsic difference in resolution between directions aligned with vs. orthogonal to gravity. These findings suggest that locomotor affordances in the environment, of which gravity is one modulator, have an effect on encoding structure and accuracy of the spatial map. This may have implications for spatial mapping not just in vertical space but in any space in which locomotion is difficult or interrupted.

## Results

### Rats explored the lattice maze fully, but adopted a layer strategy.
Rats explored the lattice mazes (Fig. 1) fully, with slightly more coverage in the aligned than the tilted configuration (Supplementary

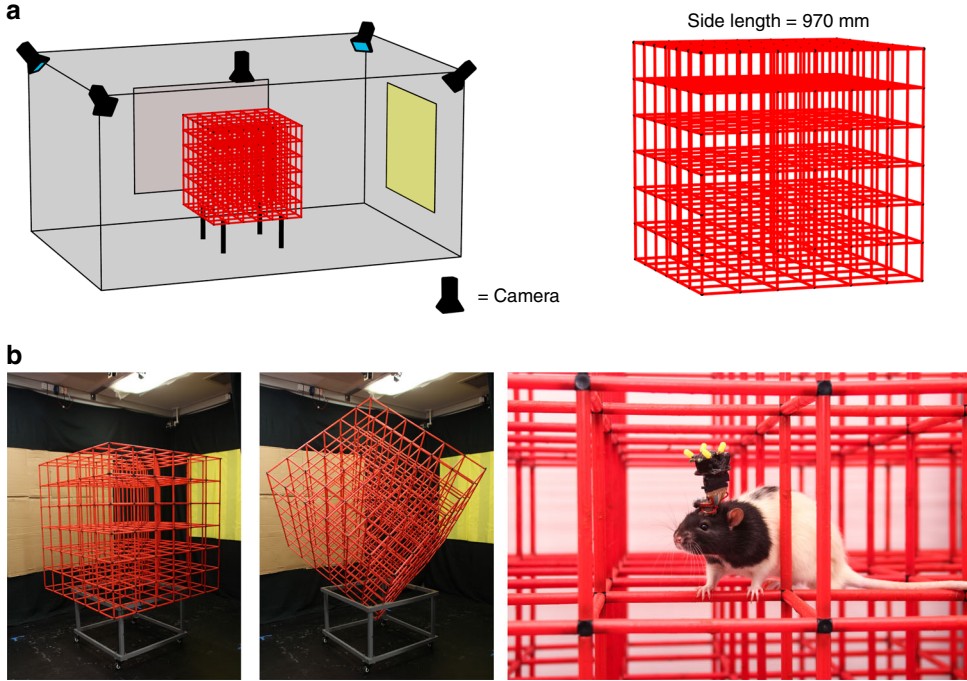

**Fig. 1 The recording room and apparatus. a** Room and maze schematic, shown in aligned configuration. **b** Photographs showing the aligned lattice maze in position for recording (left) the tilted lattice maze in position for recording (middle) and a rat implanted with an Axona microdrive exploring the aligned lattice whilst connected to the wireless headstage (right).

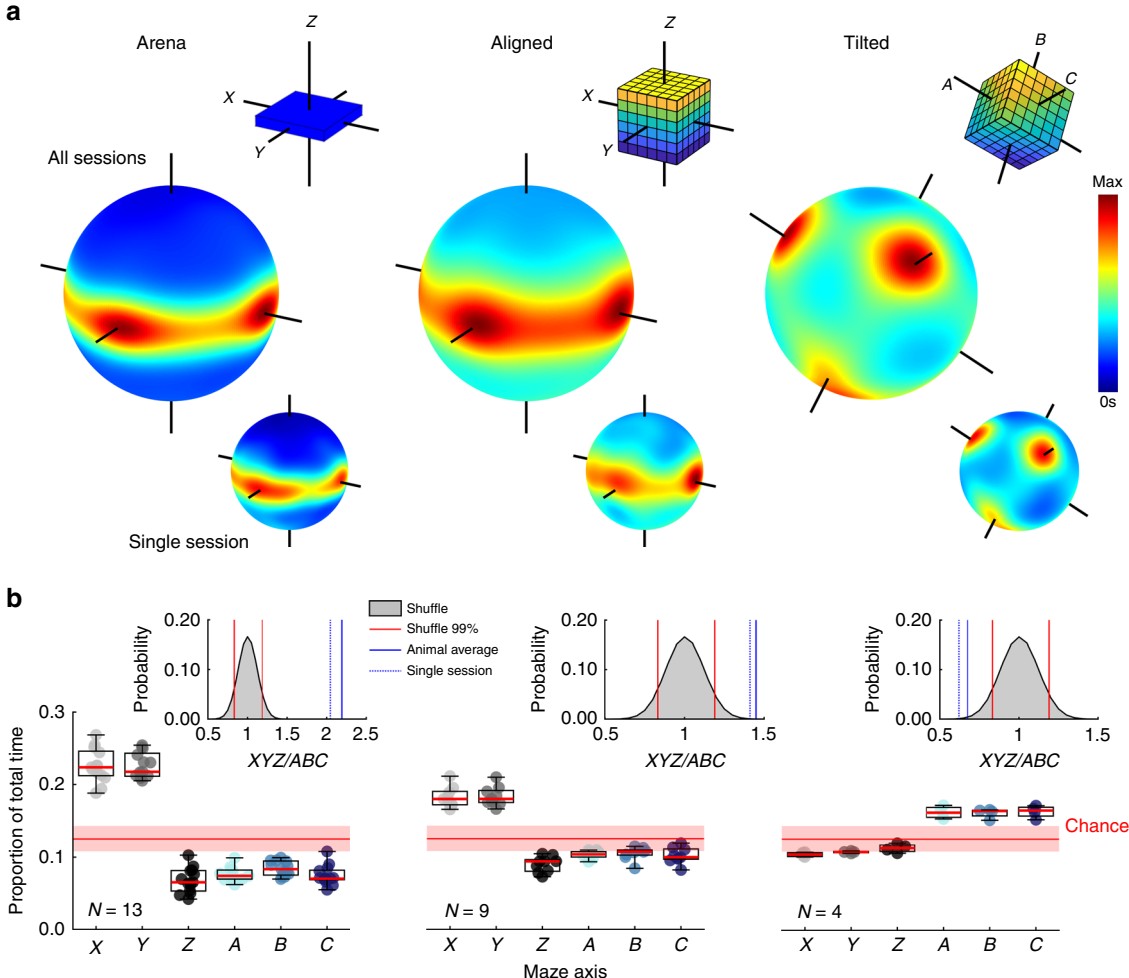

**Fig. 2 Animals moved parallel to the maze axes.** Statistical test results can be seen in Table 1. Source data are provided as a Source Data file. **a** For each maze: a schematic of the maze configuration color-coded to show height (top right inset); three-dimensional heat plot of heading direction distribution for all sessions combined (middle); same three-dimensional heat plot for a single session (bottom right inset). Note concentration around horizontal trajectories for the arena and aligned maze, and along the three axes for the tilted maze. **b** Each marker represents an animal: graphs show proportion of total time spent moving roughly parallel to each possible maze axis. Red lines show the 1st, 50th, and 99th percentile of a shuffle distribution. Inset plots show the result of a shuffle testing the probability of observing this ratio of total *XYZ* time to total *ABC* time by chance. Red lines denote the 1st and 99th percentile rank positions in the shuffled distribution of ratio values (gray area), blue line denotes the overall ratio value averaged across rats, and blue dotted line denotes the ratio value observed in the single session shown in **b**.

Fig. 1a, b). In both configurations they spent more time in the lower half and remained closer to the maze boundaries (Supplementary Fig. 1c, d). Movement speed profiles did not differ between environments (Supplementary Fig. 1f, Supplementary Data: *Movement patterns in the lattice mazes*).

In the arena and aligned lattice animals mostly moved parallel to the horizontal *X* and *Y*-axes with no preference between them (Fig. 2a, b, Table 1). They did not move along any other axes more than would be expected by chance (Fig. 2b red area). Additionally, in the aligned lattice animals moved vertically at a much slower speed compared to *X* or *Y*, confirming a strong horizontal bias in their movements[11] (Supplementary Fig. 1e, Supplementary Data: *Movement patterns in the lattice mazes*, Supplementary Movie 3). In the tilted lattice animals moved mostly parallel to the now rotated maze axes, which we referred to as *A*, *B* and *C* (Fig. 2a, b) and they did not move along any other axes more than would be expected by chance (Fig. 2b red area). The *A*, *B* and *C* axes were explored at an equal rate (Table 1) and speed (Supplementary Fig. 1e) suggesting that the animals did not have a bias for a specific axis in this configuration.

**Place fields represented lattice mazes uniformly**. In total we recorded 756 place cells in the lattice maze environments from 13 rats (Table S1). Representative place cells can be seen in Fig. 3, Supplementary Figs. 2 and 3 and Supplementary Movies 1 and 2. Cells were stable throughout the lattice maze recordings (Supplementary Fig. 4, Supplementary Data: *Recording stability*). Because we 3D-tracked rats in all three apparatuses, for comparative analyses we treated the arena as a shallow volume. The proportion of cells with at least one place field did not differ between the mazes (arena, aligned & tilted: 82.5, 85.2 and 83.8%, $\chi^2(1) = 2.49$, $p = 0.29$, CST) but these cells exhibited significantly more fields in the lattice mazes (Fig. 4a–c, Table 1). However, the number of fields expressed per cell did not scale with the volume of the mazes, resulting in fewer fields per m³ in the lattice mazes (Fig. 4d, Table 1). Instead, place field volume was larger in the aligned lattice than the arena and larger again in the tilted lattice (Fig. 4e, Table 1). However, place field diameter varied very little between mazes with only a small, albeit significant, difference between the arena and tilted lattice (Fig. 4f, Table 1). Fields were distributed throughout the lattices uniformly and in each case the

**Table 1 Statistical test results.**

| Comparison | Test | Results | Fig. |
|---|---|---|---|
| Proportion of total time along $X$ & $Y$, arena | FT | $\chi^2(1) = 0.08$, $p = 0.78$, $\eta_p^2 = 0.001$ | Fig. 2b |
| Proportion of total time along $X$ & $Y$, aligned | | $\chi^2(1) = 1.00$, $p = 0.31$, $\eta_p^2 = 0.037$ | Fig. 2b |
| Proportion of total time along $A$, $B$ & $C$, tilted | | $\chi^2(2) = 1.50$, $p = 0.47$, $\eta_p^2 = 0.125$ | Fig. 2b |
| Fields per cell, arena, aligned & tilted | KW | $\chi^2(2) = 83.60$, $p < 0.0001$, $\eta_p^2 = 0.062$ | Fig. 4b |
| Fields per m³, arena, aligned & tilted | | $\chi^2(2) = 395.49$, $p < 0.0001$, $\eta_p^2 = 0.297$ | Fig. 4c |
| Field volume, arena, aligned & tilted | | $\chi^2(2) = 63.10$, $p < 0.0001$, $\eta_p^2 = 0.037$ | Fig. 4d |
| Field diameter, arena, aligned & tilted | | $\chi^2(2) = 7.30$, $p = 0.026$, $\eta_p^2 = 0.004$ | Fig. 4e |
| Field elongation, arena, aligned & tilted | | $\chi^2(2) = 65.60$, $p < 0.0001$, $\eta_p^2 = 0.039$ | Fig. 6b |
| Field elongation arena | WSR (compare to 1) | $Z = 22.80$, $p < 0.0001$, U3 = 0 | Fig. 6b |
| Field elongation aligned | | $Z = 21.29$, $p < 0.0001$, U3 = 0 | Fig. 6b |
| Field elongation tilted | | $Z = 17.29$, $p < 0.0001$, U3 = 0 | Fig. 6b |
| Field sphericity, arena, aligned & tilted | KW | $\chi^2(2) = 426.5$, $p < 0.0001$, $\eta_p^2 = 0.251$ | Fig. 6c |
| Field sphericity arena | WSR (compare to 1) | $Z = -22.80$, $p < 0.0001$, U3 = 1 | Fig. 6c |
| Field sphericity aligned | | $Z = -21.29$, $p < 0.0001$, U3 = 1 | Fig. 6c |
| Field sphericity tilted | | $Z = -17.29$, $p < 0.0001$, U3 = 1 | Fig. 6c |
| Field length distributions, aligned | Multiple KS with Bonferroni | $X$ vs $Y$: $z = 0.05$, $p > 0.99$ | Fig. 6d |
| | | $X$ vs $Z$: $z = 0.09$, $p = 0.03$ | |
| | | $Y$ vs $Z$: $z = 0.11$, $p = 0.003$ | |
| Field length distributions, tilted | | $p > 0.2$ in all cases | Fig. 6d |
| Autocorrelation aligned, $X$, $Y$ & $Z$ | FT | $\chi^2(2) = 128.9$, $p < 0.0001$, $\eta_p^2 = 0.10$ | Fig. 8a |
| | | $X$ vs $Z$ & $Y$ vs $Z$, $p < 0.0001$, $X$ vs $Y$, $p > 0.99$ | |
| Autocorrelation tilted, $A$, $B$ & $C$ | | $\chi^2(2) = 1.4$, $p = 0.49$, $\eta_p^2 = 0.002$ | Fig. 8a |
| Proportion of spatial information aligned, $X$, $Y$ & $Z$ | | $\chi^2(2) = 153.3$, $p < 0.0001$, $\eta_p^2 = 0.119$ | |
| | | $X$ vs $Z$ & $Y$ vs $Z$, $p < 0.0001$, $X$ vs $Y$, $p > 0.99$ | Fig. 8b |
| Proportion of spatial information tilted, $A$, $B$ & $C$ | | $\chi^2(2) = 1.4$, $p = 0.498$, $\eta_p^2 = 0.001$ | |
| Area under curve, aligned, $X$, $Y$ & $Z$ | | $\chi^2(2) = 9.1$, $p = 0.011$, $\eta_p^2 = 0.005$ | |
| | | $X$ vs $Y$, $p > 0.99$, $X$ vs $Z$, $p = 0.047$, $Y$ vs $Z$, $p = 0.017$ | Fig. 8c |
| Area under curve, tilted, $X$, $Y$ & $Z$ axes | | $\chi^2(2) = 4.7$, $p = 0.094$, $\eta_p^2 = 0.0039$ | |
| Area under curve, tilted, $A$, $B$ & $C$ axes | | $\chi^2(2) = 2.8$, $p = 0.25$, $\eta_p^2 = 0.0024$ | |
| | | Median $A$, $B$ & $C$: 4.74, 4.74. 4.51 | – |

Test abbreviations and details can be found in Methods: *Statistics*.

median field centroids lay close to the maze center (Fig. 5). There was no significant relationship between the numbers of fields expressed in the arena and the lattice mazes (Supplementary Fig. 10, Supplementary Data: *Comparison of firing properties between mazes*).

**Place fields were elongated rather than spherical**. Place fields took on different shapes in the mazes; most fields were elongated in the lattice mazes while they exhibited a flattened shape in the arena (Supplementary Fig. 5). Only a minority of fields in each maze were isotropic or more spherical than would be expected by chance (Fig. 6b, c text percentages). Instead, place fields in all conditions were slightly elongated, with elongation indices and sphericity that deviated significantly from 1 (Fig. 6b, c, Table 1). It is unlikely these effects were due to inhomogeneous sampling (Supplementary Fig. 6). In the aligned lattice the distribution of field heights (length along Z) deviated from the distributions of length along X or Y (Table 1) and had a significant bimodal appearance (Supplementary Fig. 7a, Supplementary Data: *Field elongation)*. By contrast, in the tilted maze all axes shared a similar unimodal distribution (Fig. 6d). Place field elongation in the lattice mazes was weakly but significantly positively correlated with the distance of the field from the maze center, but we found no relationship between elongation and experience or cluster quality in any of the mazes (Supplementary Fig. 7b–d). When cells had a field in both the arena and aligned lattice there was no relationship between their elongation in the two configurations, although there was a weakly negative correlation between the arena and tilted lattice data; when cells had multiple fields in the

lattice mazes their lengths were not more similar than would be expected by chance (Supplementary Fig. 10b, c, Supplementary Data: *Comparison of firing properties between mazes*).

**Place fields were elongated parallel to the maze axes**. Given that the majority of fields were elongated instead of spherical we investigated whether they were elongated along a common orientation (Fig. 7a; Supplementary Methods: *Field orientation and size*). In the arena the 3D orientations of place fields were not random; instead the majority of fields had their longest axis running parallel to either the X or Y axis, parallel to the walls of the arena (Fig. 7b). A shuffle analysis revealed that more fields were oriented along these axes than would be expected by chance; this was not true of any other axis (Fig. 7c red shaded area). These two axes shared a similar number of fields (Fig. 7c, overlap with confidence intervals) and these fields all had a similar length (median X and Y length, 68.46 and 66.96 cm, $\chi^2(1) = 0.01$, $p = 0.97$, $\eta_p^2 < 0.0001$, K-W).

In the aligned lattice the majority of fields had their longest axis running parallel to either the $X$, $Y$ or $Z$ axes which were also parallel to the lattice bars (Fig. 7b). Again, a shuffle analysis revealed that these orientations were the only ones with more fields than expected by chance (Fig. 7c red shaded area). These shared a similar number of fields (Fig. 7c, overlap with confidence intervals) but in this case the fields aligned with the $Z$ axis were significantly longer (median length, $X$, $Y$ and $Z$: 64.24, 57.57 & 78.16 cm, $\chi^2(2) = 26.8$, $p < 0.0001$, $\eta_p^2 = 0.055$, K-W, $X$ vs $Y$, $p = 0.074$, $X$ vs $Z$, $p < 0.0085$, $Y$ vs $Z$, $p < 0.0001$).

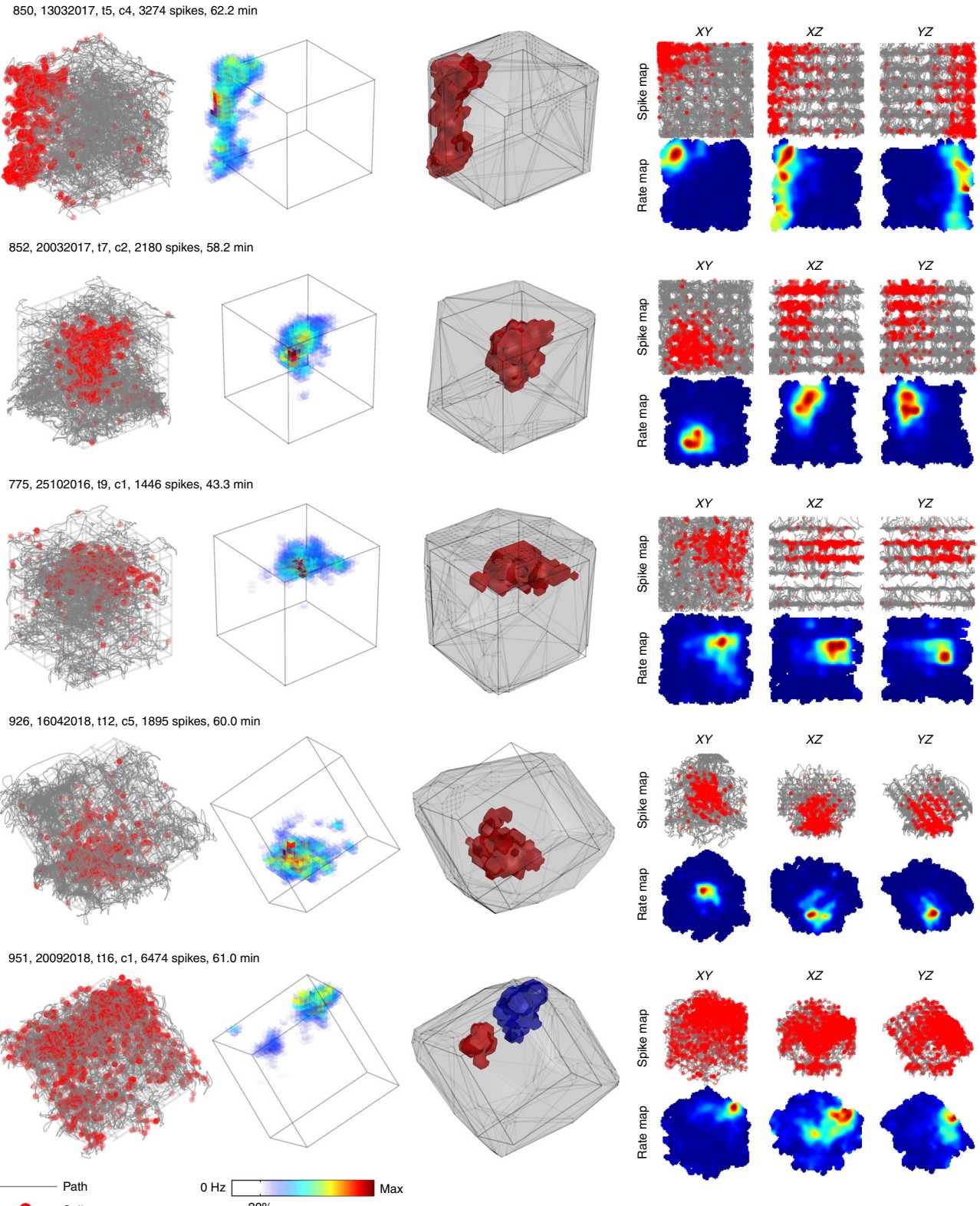

**Fig. 3 Representative example place cells and their activity in the lattice mazes.** For additional examples see Supplementary Figs. 3 and 4 and Supplementary Movies 1 and 2. Five cells are shown, one per row. First column shows the path of the animal and spikes plotted as red markers. Second column shows the three-dimensional firing rate map. Colors denote firing rate and areas of low or no firing are transparent. Third column shows the convex hull of the dwell time map as a gray outline and the convex hull of any detected place field(s) as separate (color-coded) polygons. Last column shows the spike and firing rate maps when the data are projected onto the three possible cardinal planes.

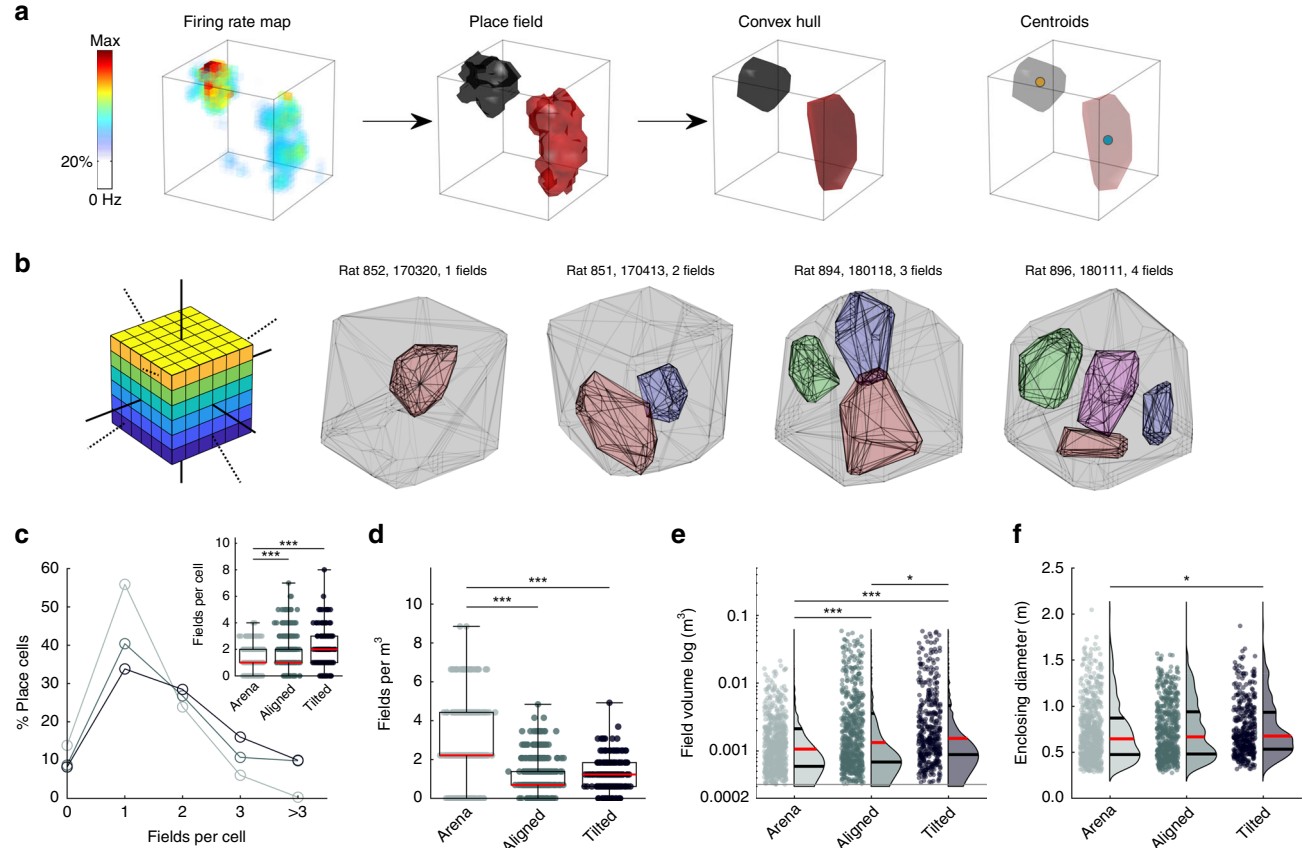

**Fig. 4 Fields were less numerous in the lattice mazes but larger in volume.** Source data are provided as a Source Data file. Markers in boxplots represent fields. Omnibus test results can be seen in Table 1; post hoc test results are displayed here: *significant at the 0.05 level, **significant at the 0.01 level, ***significant at the 0.001 level. **a** Schematic demonstrating the process of place field detection and analysis. An example firing rate map (left) is thresholded at 20% of the peak firing rate and regions which passed our criteria were considered place fields (second plot). We can visualize these regions as convex hulls (third plot) and extract features such as their centroid (right plot). **b** Aligned lattice maze schematic and place field convex hulls of four example place cells exhibiting 1–4 place fields. **c** Number of place fields exhibited by place cells in each maze. Inset: same data in boxplot representation. **d** Number of fields per cubic meter exhibited by place cells in each maze. **e** Distribution of place field volumes observed in each maze. **f** Enclosing diameter of place fields in each maze.

In the tilted lattice a different pattern of results emerged; fields were mainly oriented parallel to the A, B and C axes which were also parallel to the (now rotated) lattice bars (Fig. 7b). As before, these orientations were the only ones with more fields than chance (Fig. 7c red shaded area); they shared an equal proportion of fields (Fig. 7c, overlap with confidence intervals) and a similar length (median length, A, B and C: 69.13, 73.43 and 61.84 cm, $\chi^2$ (2) = 0.80, p = 0.68, $\eta_p^2$ = 0.002, K-W). For all three mazes an independent approach confirmed that field elongation was best described as parallel to each maze's axes (Supplementary Fig. 8, Supplementary Data: *Field orientation*). Lastly, when cells had multiple fields in the lattice mazes their orientations were not more similar than would be expected by chance (Supplementary Fig. 10d, Supplementary Data: *Comparison of firing properties between mazes*).

**Vertical spatial coding was less accurate**. If fields were larger along a specific dimension, firing rate maps would be more highly autocorrelated along this dimension and the spatial information content conveyed by the cell's activity would be lower (see Supplementary Fig. 9a for an example). This was the case for the Z-axis of the aligned lattice (Fig. 8a, b, Table 1) but not for the tilted lattice where there were no such differences

between the A, B and C axes (Fig. 8a, b, Table 1). Similar effects were also observed using a variety of other measures (Supplementary Fig. 9c–f; Supplementary Data: *Autocorrelation, spatial information and binary morphology*). Down-sampling trajectory data to account for the biases in animals' movements confirmed that these biases do not account for the effects described here (Supplementary Fig. 6; Supplementary Data: *Trajectory downsampling*).

To investigate this reduced vertical resolution at the level of individual place fields we projected fields onto three orthogonal axes and calculated the area under the curve (AUC) for each. Place field AUCs in the aligned lattice were significantly larger along the Z dimension when compared to X and Y (Fig. 8c, Table 1). In contrast, there was no significant difference in the tilted lattice when comparing the X, Y and Z axes or A, B and C axes (Fig. 8c, Table 1). Place field firing rate curves and the results of a similar but independent approach can be seen in Supplementary data (Supplementary Fig. 9e, f, Supplementary Data: *Autocorrelation, spatial information and binary morphology*).

We investigated whether the reduced spatial specificity along the vertical dimension may be due to instability of the cells in this dimension over time (Supplementary Methods: *Field stability*). Correlations between the first and second half of each session were significantly higher than chance in all cases;

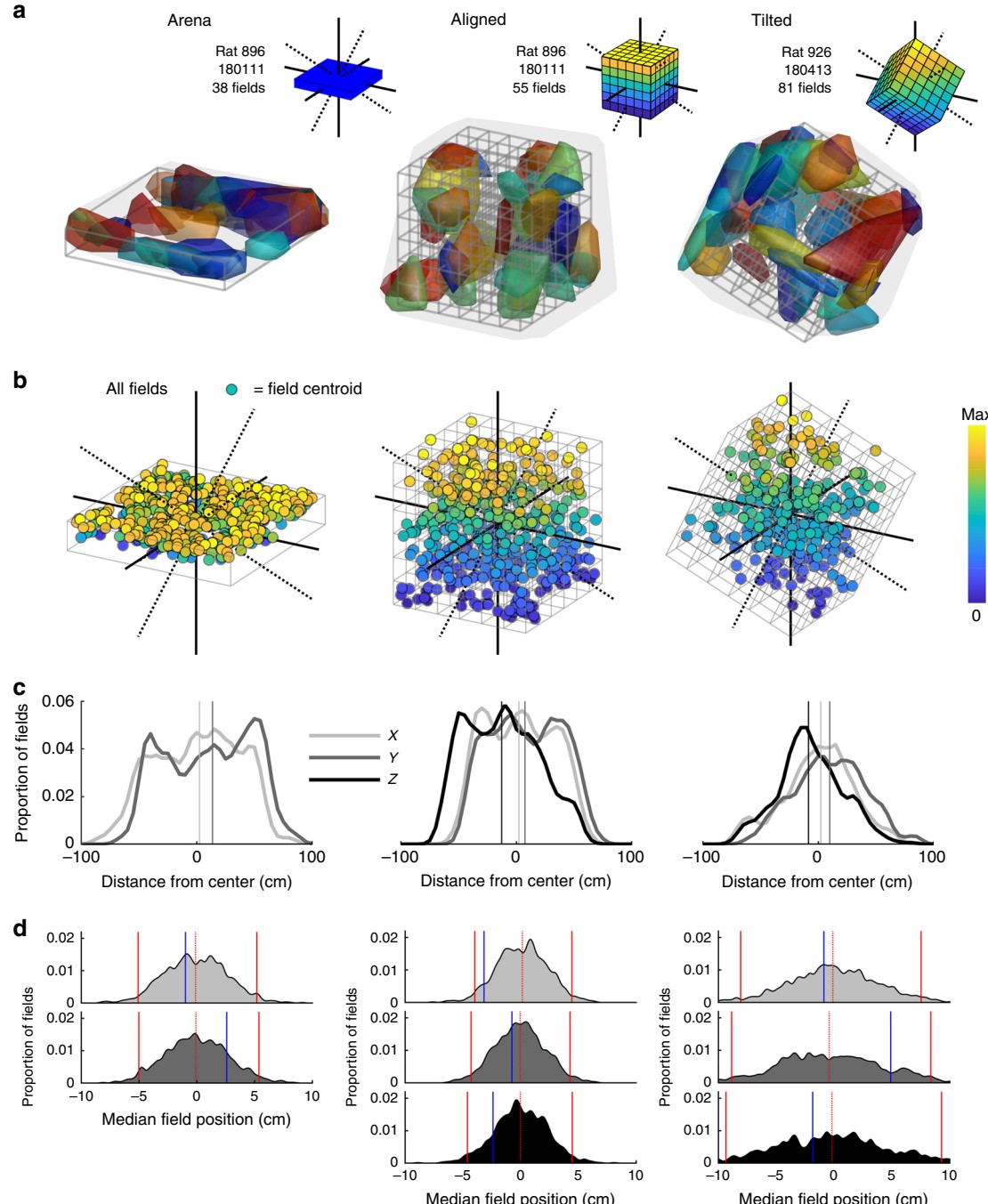

**Fig. 5 Place fields were distributed throughout the three mazes.** Source data are provided as a Source Data file. **a** Representative arena, aligned and tilted lattice recording sessions demonstrating homogenous distribution of fields. To allow clearer distinction of separate (color-coded) fields, only fields with a volume less than 300 voxels (one voxel = 50 mm cube) are shown (~2/3 total). Inset schematics show the maze orientation and axes. **b** Location of all recorded place field centroids. Colors denote vertical position. **c** Columns follow the mazes as above; the kernel smoothed distribution of place fields along the X, Y and Z dimensions of each maze (Z is not shown for the arena) relative to the maze center. Vertical lines represent the median value of these distributions. **d** Columns follow the mazes as above; from top to bottom graphs show the results of a shuffle analysis on the distribution of fields along the X, Y and Z axes (Z is not shown for the arena). Blue lines denote the median position of real place fields along these axes relative to the maze center. Shaded areas represent the distribution of median values obtained from 1000 shuffles. Red lines show the 2.5 and 97.5 percentile rank positions in the shuffled distributions, red dotted lines denote the 50th percentile rank or median of the shuffle distributions.

furthermore, in the majority of cases the median correlation value exceeded the 95[th] percentile of the shuffled distribution. However, place cell activity was generally most stable when projected onto the XY plane, suggesting that cells were indeed less stable in the vertical dimension (Supplementary Data: *Field*

*stability*). Stability was generally lower in the tilted lattice comparisons, which is also in agreement with reduced spatial information and increased sparsity in this maze (Supplementary Fig. 10a, Supplementary Data: *Comparison of firing properties between mazes*).

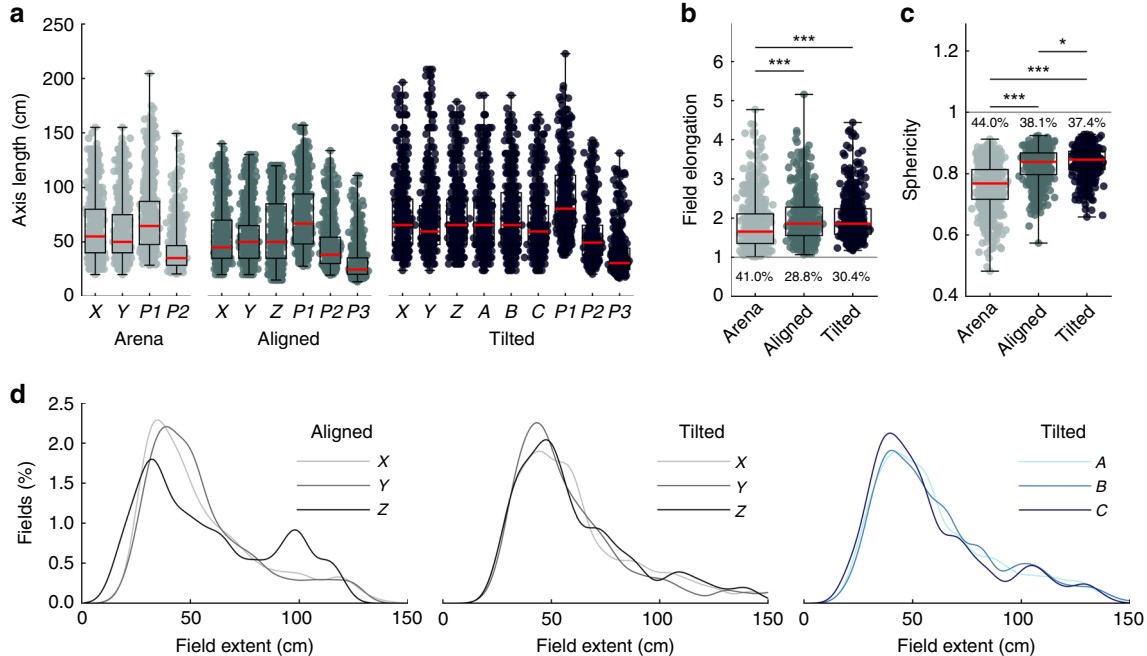

**Fig. 6 The majority of place fields were elongated instead of spherical.** Source data are provided as a Source Data file. Markers represent place fields. Omnibus test results can be seen in Table 1, post hoc test results are displayed here: *significant at the 0.05 level, **significant at the 0.01 level, ***significant at the 0.001 level. **a** Cartesian axis and principal axis lengths of place fields in all three mazes (P1-3 are the principal, semi-major and semi-minor axes respectively). **b** Elongation index of all place fields in each maze. An index of 1 (gray line) would indicate a spherical field, higher values indicate elongation. The percentage of fields that are equally or more spherical than would be expected by chance is given by the text below each boxplot. **c** Same as **b** but for sphericity. Values <1 indicate deviation away from a sphere. **d** Probability density functions of the *XYZ* and *ABC* field length data in **a**. The distribution of *Z* lengths differs significantly from *X* and *Y* in the aligned lattice and the distribution of *Z* lengths appears to be bimodal (Supplementary Data: Field elongation). In the tilted lattice there are no differences between *X*, *Y* & *Z* or *A*, *B* & *C*. See Supplementary Fig. 7 for further field elongation analyses.

## Discussion

This experiment investigated how hippocampal place cells represent three-dimensional, volumetric space in rats, which are predominantly surface-traveling animals. The aim was to see whether all three dimensions would be represented equally, as they are in freely flying bats, implying an isotropic and volumetric map of space. We used three-dimensional lattice environments where the rats were free to move in any direction, restricted only by the underlying structure of the environment. In one setting the lattice structure was aligned with gravity and in the other it was tilted, enabling us to disentangle restrictions due to gravity from restrictions due to maze structure. We found that place fields packed the lattice space with ovoid fields, in a similar manner to bats, indicating a volumetric map. However the fields were slightly elongated along the maze axes. This was more pronounced in the vertical dimension for the aligned lattice, indicating an interaction between the effects of structure and gravity on place fields. Taken together with previous findings, this suggests that the hippocampal map of three-dimensional space is not fixed but is flexibly shaped by environment structure, perhaps via the movement constraints/affordances it provides. Below, we discuss the findings that lead to this conclusion, and its implications.

When the lattice was aligned with gravity we found that rats explored using a "layer strategy" in which they fully explored one level before moving to the next, meaning far fewer vertical movements than horizontal ones—this replicates previous findings and is consistent with the notion that animals will execute the easier parts of a multi-stage journey first[11]. When the maze was tilted, all three principal axes became sloped relative to gravity and thus equally easy/hard to traverse, and the layer strategy disappeared. However, we also found that rats spent more time in the lower part of the mazes.

In both maze alignments, we found that place fields were distributed evenly throughout the volume of the lattices and had broadly similar properties in vertical vs. horizontal dimensions (were isotropic). The volumes of fields in the lattice that we observed were smaller than those predicted by extrapolating from our observations in the arena. This is consistent with the fact that although place fields had similar diameters in the three mazes, the number of fields exhibited per cell did not change between environments, resulting in a significantly lower density of fields per m³ in the lattice mazes. This finding is in contrast to findings in 2D suggesting that place cells exhibit more fields in larger environments[12].

We next looked at the structure of place fields in the different dimensions, finding that place fields tended to be elongated, as has been generally seen in two dimensions[6,13,14]. Elongation did not occur in every direction but was almost always in the direction of the maze axes/boundaries. Two related explanations for why this might occur present themselves. One is that the maze boundaries, represented by the termination of the cross-bars, serve to anchor place fields in a similar way to walls and edges in a flat environment, possibly via boundary cells found in the subiculum[15] and medial entorhinal cortex[16]. These have been shown to respond to both walls and drops[17,18] and are able to "reset" the spatial firing of entorhinal grid cells[19]. Since the effect of anchoring falls off with distance due to accumulating path integration error, fields should tend to be narrower in the direction orthogonal to the nearest boundary, for which distance to the wall is small, and elongated in the direction that runs between the two more distant boundaries.

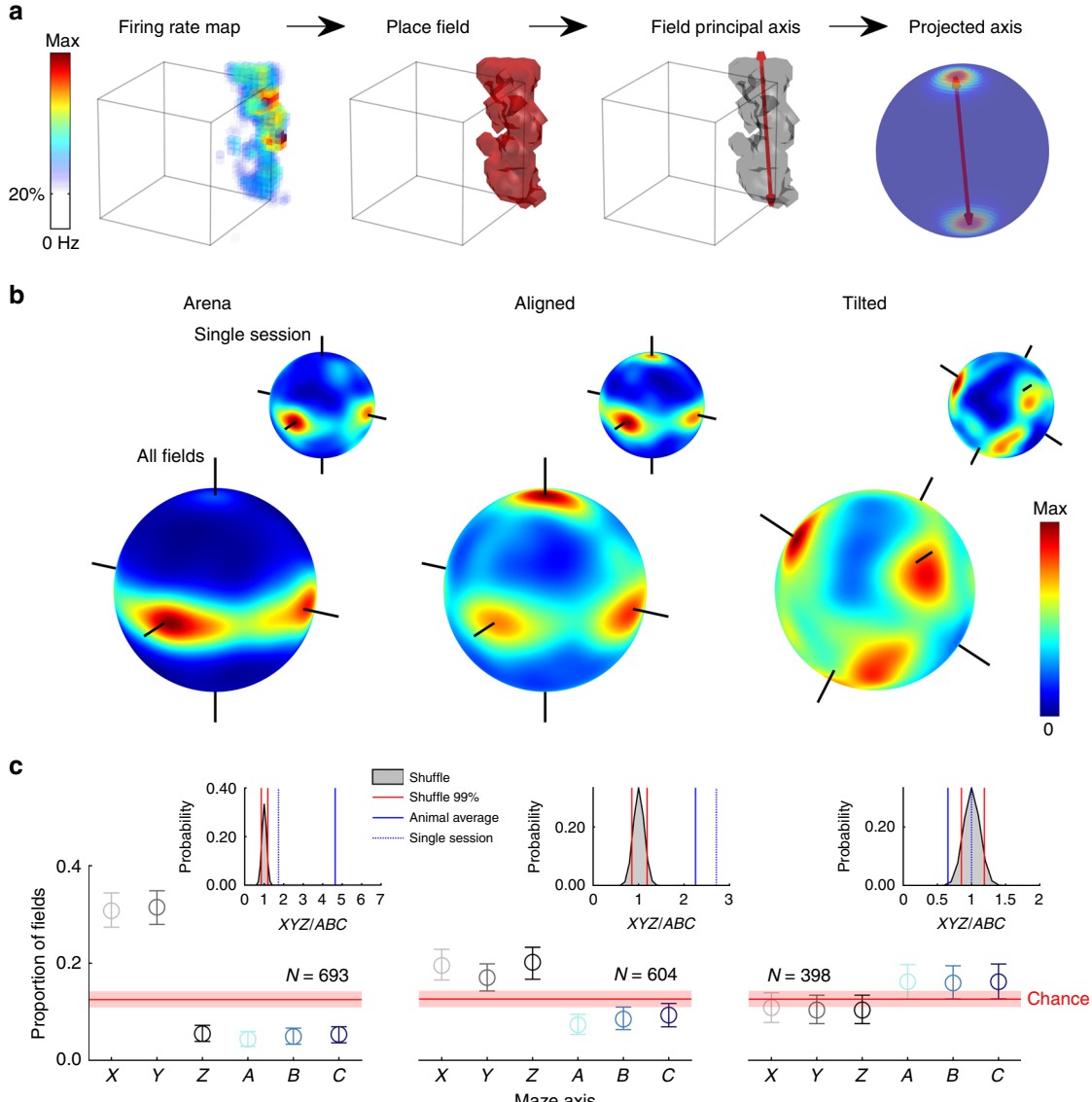

**Fig. 7 Place fields were oriented parallel to the maze axes.** Source data are provided as a Source Data file. **a** Schematic showing how the orientation of place fields was extracted and visualized. A place field is detected in an example firing rate map through thresholding, the principal or longest axis of this field can then be extracted (red arrows, third plot). To visualize the orientation of multiple fields we project these axes onto a unit sphere and generate a spherical Von-Mises kernel smoothed density map, where hot colors denote that many fields 'pointed' their principal axis in this orientation. **b** Three-dimensional heat plots of place field orientation for the three maze configurations; inset heat plots (top right) show data for a single session, large plots (bottom) show data for all place fields. Note concentration around the three axes of the aligned and tilted mazes. Flat cylindrical projections can be seen in Supplementary Fig. 8. **c** Graphs show proportion of total fields oriented roughly parallel to each possible maze axis. Circles give the observed proportion per axis, error bars represent 95% confidence intervals calculated through a bootstrapping procedure. Red lines show the 50th percentile of a shuffle distribution while shaded red areas denote the interval between the 2.5th and 97.5th percentiles. Inset plots show the result of a shuffle testing the probability of observing this ratio of total *XYZ* fields to total *ABC* fields by chance. Red lines denote the 1st and 99th percentile rank positions in the shuffled distribution of ratio values (gray area), blue line denotes the overall ratio value averaged across rats, and blue dotted line denotes the ratio value observed in the single session shown in **c**.

The other explanation is that perhaps fields tend to be elongated in the direction more frequently traversed by the animals, or that is traversed for a longer continuous time. Since animals can spend relatively little time running directly towards or away from a boundary, but much time running back and forth along it, synaptic plasticity would have more opportunity to "grow" fields along the direction of travel[20,21]. A similar argument could explain elongation along maze axes, except that rats rarely moved vertically in the aligned lattice yet fields were still elongated along this axis. In the present experiment we did not investigate this further by rotating the axes relative to the

boundaries, but this would be an interesting direction for future experiments.

We next looked at whether field elongation was greater in the vertical dimension. Previous research in rats on vertical surfaces found the vertical dimension to be represented differently to horizontal dimensions, although the exact nature of this difference depended on the movement patterns. When the rats climbed on pegs but remained oriented mainly horizontally then place fields were elongated vertically[6], whereas when the animals climbed by clinging to chicken wire and were thus aligned with the wall then place fields were sparser, but no longer vertically

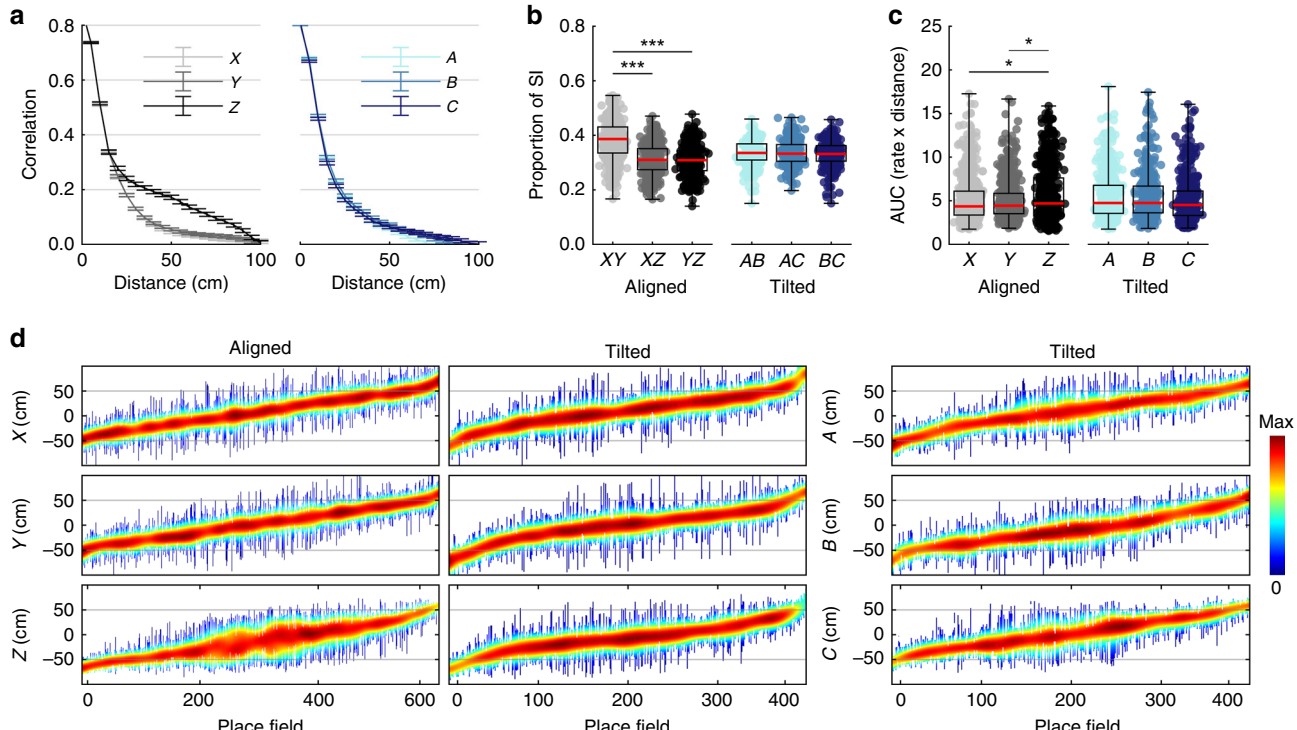

**Fig. 8 Spatial information was lowest along the Z dimension of the aligned lattice.** Source data are provided as a Source Data file. Omnibus test results can be seen in Table 1; post hoc test results are displayed here: *significant at the 0.05 level, **significant at the 0.01 level, ***significant at the 0.001 level. **a** Mean and SEM autocorrelation found for all place cells in the aligned lattice (left) and tilted lattice (right) at increasing distances or spatial lag. **b** Proportion of total spatial information found per projection for the aligned lattice (left) and tilted lattice (right). **c** Area under the firing rate curve (AUC) produced when place fields are projected onto each axis for both lattice mazes. **d** Span and normalized activity of every recorded place field in the lattice mazes. Vertical lines represent place fields, ordered along the x-axis by their position relative to the lattice central node. Line color represents normalized firing rate.

elongated[7]. In a study of flying bats, fields were not different from spherical[10]. In the present experiment we found an increase in place field elongation in the vertical dimension, which was also represented less stably: however this was only when the maze was aligned with gravity, and not when it was tilted. The aligned configuration is the one that induced differential movement patterns, with freer movement in *x-y* than in *z*. Putting all these experiments together, the hypothesis emerges that place fields have less resolution in a dimension in which the animal does not freely travel in the direction of its body axis. This might occur if the distance-tracking process is not uniform in all directions but works best in the direction of travel.

The heterogeneity of findings in the different mazes points to a fundamental conclusion which is that there is not a fundamental, holistic map of space that permeates three-dimensional space and is sampled by the animal as it moves through the space over various surfaces. This is because no unitary map structure could account for field elongation on the pegboard, field sparsity on the chicken-wire "cling wall" and in the volumetric mazes. Rather, it seems that a different kind of place cell representation is recruited depending on environment structure and perhaps task demands.

Our findings of a volumetric place cell map agree not only with the data from bats but also from recent neuroimaging work in humans, suggesting the encoding resolution for movement along a vertical axis in a lattice maze does not differ greatly from horizontal[22]. However behavioral experiments suggest a subtle difference, with an advantage in memory for horizontal as compared to vertical space[4]. More recent evidence suggests that people wrongly estimate the position of objects in a well-known building, giving the overall effect of a vertically elongated but

horizontally contracted spatial representation[23] which is in agreement with our finding of increased elongation along the vertical dimension (but see ref. [24]). The path which participants use to explore a building has also been shown to play a crucial role; people who explore a building by mainly vertical paths were better at recalling the positions of vertically arranged objects than people who explored the same building by mainly horizontal paths[25] supporting the importance of environment affordances in the development of spatial representations.

In this paper we have shown that surface-dwelling animals such as rats do have a neural volumetric representation of space and that this representation exhibits many of the same characteristics as two-dimensional representations. Place fields are elongated parallel to the primary axes of every environment with a slight bias towards vertical elongation, and spatial coding and stability are significantly reduced along this dimension, suggesting that these animals may not encode the vertical dimension with equal accuracy. Future research will need to investigate these effects in volumetric animals such as flying bats to determine if spatial maps share a common organization across species or if separate neural mechanisms exist in volumetric animals. Our results point to an important effect of environmental affordances, evidence of which can be seen in other spatial mapping literature. However, more research is needed to tease apart the relationship between affordances, geometry, gravity and behavioral sampling. This could look to combine recordings with behavioral training, to increase sampling of the more difficult vertical dimension. Our results, combined with those from recent experiments on the head direction system[26,27] suggest that the rodent spatial navigation network may be far better at mapping three-dimensional

**Table 2 Summary of statistical tests, their abbreviations and software used.**

| Parametric Yes/No | Test | Abbreviation | Effect size | Software |
|---|---|---|---|---|
| N | Wilcoxon rank sum | WRS | Cohen's U3 (U3, fraction of values in group 1 less than those in group 2 or the test value in a one-sample test) | Matlab function *ranksum* |
| N | Wilcoxon signed rank | WSR | – | Matlab function *signrank* |
| N | Kolmogorov–Smirnov | KS | – | Matlab kstest2 |
| N | Chi-square test of expected proportions | CST | Odds ratio (OR, the ratio of the odds of an event occurring in one group to the odds of it occurring in another group) | GraphPad, QuickCalcs |
| Y | Bootstrap modality test | – | – | Matlab function, *bootmode* |
| N | Kruskal–Wallis | KW | Partial eta squared ($\eta_p^2$, proportion of variance in the dependent variable explained by an independent variable) | Matlab function *kruskalwallis* |
| N | Friedman test | FT | | Matlab function *friedman* |
| Y | Univariate ANOVA | – | – | SPSS 25, generalized linear models |
| Y | Repeated measures ANOVA | – | – | |
| N | Permutation *F*-test | – | – | Matlab, custom functions |

space than previously thought. This confirms the relevance of rodents such as rats in studying these representations and raises questions regarding three-dimensional encoding of other spatial cells such as grid and boundary cells. Overall the strong spatial representations we have observed in place cells points to the possibility of distinct spatial representations of volumetric space by grid cells which have yet to be explored.

## Methods

**Statistics and figures**. If data were found to deviate significantly from a normal distribution (Matlab functions *lillietest*, *skewness*, *kurtosis*) we used non-parametric tests, and post hoc tests compared average ranks (Matlab function *multcompare*, Bonferroni correction). Otherwise, we used parametric tests, and post hoc tests compared population means (Matlab function *multcompare*, Bonferroni correction). In the case of multivariate comparisons, where we sought to determine any interaction effects, we employed generalized linear models using SPSS. Where possible we report effect sizes for each test. Unless otherwise stated all statistical tests are two-tailed. Table 2 gives a summary of the tests used, how they were conducted and how they are reported in the text. In all figures * = significant at the 0.05 level, ** = significant at the 0.01 level, *** = significant at the 0.001 level. For all box plots, red lines denote the sample median, boxes denote interquartile range, whiskers span the full range of the data and markers represent individual data points. Similar results to those reported in main text were also observed when only analyzing one session per animal (the session with the most place cells; Supplementary Data: *Subsampled dataset analysis*, Supplementary Fig. 11, Tables S2 and S3).

**Animals**. Thirteen animals were used for single unit electrophysiological recording (nine in the lattice, four in the diagonal lattice), at which point they weighed approximately 400–450 g. Prior to surgery all animals were housed for a minimum of 8 weeks in a large (2.15 m × 1.55 m × 2 m) cage enclosure, lined on the inside with chicken wire. This was to provide the rats with sufficient experience of climbing in a three-dimensional environment. During this time they were given unlimited access to a miniature version of the lattice maze. This was composed of similar lattice cubes (55 × 55 × 55 cm) but with a slightly smaller spacing (11 cm) and was oriented to match the experimental version appropriate to the rats (i.e., rats recorded in the aligned lattice were exposed to a miniature aligned lattice, rats recorded in the tilted lattice were exposed to a miniature tilted lattice). Animals were housed individually in cages after surgery and there they were given access to a hanging hammock or climbable nest box for continued three-dimensional experience.

The animals were maintained under a 12 h light/dark cycle and testing was performed during the light phase of this cycle. Throughout testing, rats were food restricted such that they maintained approximately 90% (and not less than 85%) of their free-feeding weight. This experiment complied with the national [Animals

(Scientific Procedures) Act, 1986, United Kingdom] and international [European Communities Council Directive of 24 November 1986 (86/609/EEC)] legislation governing the maintenance of laboratory animals and their use in scientific experiments.

**Electrodes and surgery**. A combination of Axona (MDR-xx, Axona, UK) and tripod design (Kubie, 1984) microdrives were used (rats 750, 770, 775 Kubie drives, all other rats Axona drives). Drives supported four or eight tetrodes, each of which was composed of four HML coated, 17 μm diameter, 90% platinum 10% iridium wires (California Fine Wire, Grover Beach, CA) which were gold plated (Non-Cyanide Gold Plating Solution, Neuralynx, MT) in order to reduce the impedance of the wire to a plated impedance in the range of 180–300 kΩ. Microdrives were implanted using standard stereotaxic procedures under isoflurane anesthesia[28]. Electrodes were lowered to just above the CA1 cell layer of the hippocampus (−3.5 mm AP from bregma, ±2.4 mm ML from the midline, ~1.5 mm DV from dura surface). See Supplementary Fig. 16 histology results.

**Apparatus**. A detailed description of the room and apparatus can be found in Supplementary Methods: *Apparatus*, photographs and schematics can be seen in Fig. 1. Briefly, we used three main pieces of experimental apparatus: a square open field environment ('arena'), a cubic lattice composed of horizontal and vertical climbing bars ('aligned' lattice) and the same lattice rotated to stand on one of its vertices ('tilted' lattice). Rats were recorded freely foraging in the arena for randomly dispersed flavored puffed rice (CocoPops, Kelloggs, Warrington, UK) and foraging in the lattice maze for malt paste (GimCat Malt-Soft Paste, H. von Gimborn GmbH) affixed to bars of the lattice.

**Recording setup and procedure**. A detailed description of the recording setup used can be found in Supplementary Methods: *Recording setup and procedure*. Briefly, single unit activity was observed and recorded using a custom built 64-channel recording system (Axona, St. Albans, UK) and a wireless headstage (custom 64-channel, W-series, Triangle Biosystems Int., Durham, NC) mounted with infrared LEDs. Five infrared sensitive CCTV cameras (Samsung SCB-5000P) tracked the animal's position at all times. For experimental sessions, rats were recorded for a minimum of 18 min in the arena, followed by a minimum of 45 min in one configuration of the lattice and a further minimum 16 min in the arena. Video footage of animal behavior and position tracking in the arena and aligned lattice can be seen in Supplementary Movie 4.

**Behavioral analyses**. A detailed description of positional estimation and three-dimensional trajectory reconstruction can be found in Supplementary Methods: *Trajectory reconstruction*. A detailed description of the methods used to analyze 3D behavior and heading directions can be found in Supplementary Methods: *Behavior and spherical heat maps*. Briefly, we calculated the instantaneous three-dimensional heading of the animal as the normalized change in position between time points: essentially 3D vectors pointing from one position to the next. We

projected these heading vectors onto a unit sphere and generated a heatmap of the result using a density estimation approach designed for spherical data. This process can be seen in Supplementary Movie 3. Using the underlying points we calculated the proportion of heading vectors (i.e., the proportion of total time) aligned, within a narrow range, parallel to each maze axis. By dividing position data based on the position of the lattice we were also able to calculate the time spent in the inner and outer 50% volume of the lattice or the top and bottom 50%. To compare speed profiles between mazes, for each session we calculated the proportion of time spent moving at speeds between 2 and 50 cm/s in 2 cm/s wide bins.

**Firing rate maps**. For all analyses other than those described in *Recording stability* and *Field stability*, three-dimensional volumetric firing rate maps were constructed using a similar approach to that reported previously[29]. The firing rate in each voxel ($50 \times 50 \times 50$ mm) was calculated as the distance from the voxel center to every recorded spike in the neighboring 26 voxels, divided by the distance to every position data point in these voxels. These distances were weighted using a truncated Gaussian function such that spikes and position data closer to a voxel's center had more influence on that voxel's firing rate and data outside the neighboring 26 voxels had no influence on the firing rate. The Gaussian used was defined as:

$$g(x) = e^{-0.5\left(\frac{\{x:x<d\}}{\sigma}\right)}$$

Firing rate was then calculated as:

$$f(x) = \frac{\sum_{i=1}^{n} g(S_i - x)}{\int_0^T g(y(t) - x)}$$

where $d$ is the distance threshold of the Gaussian, which was set to 1.5 voxels, $\sigma$ is the standard deviation of the Gaussian, which was set to 1 voxel width, $S_i$ represents the position of every recorded spike, $x$ is the voxel center, the period $[0T]$ is the recording session time period, $y(t)$ is the position of the rat at time $t$. If the rat did not explore within 100 mm of a voxel, or if he spent less than 1 s there, the voxel was considered unvisited.

**Recording stability**. To verify that cells were stably recorded during our maze sessions we computed the Pearson correlation between our first and second arena ratemaps (recorded before and after lattice maze sessions). These maps were generated by projecting data onto the Cartesian planes before calculating 2D firing rate maps in the standard manner; as bivariate histograms with 5 cm square bins smoothed using a Gaussian with 5 cm standard deviation (Matlab function *imgaussfilt*). Volumetric maps were generated as multivariate histograms with 5 cm cubic bins smoothed using a Gaussian with 5 cm standard deviation (Matlab function *imgaussfilt3*). In both cases spike and position data were truncated to include only the data falling within the lattice maze frame. We compared this data to values calculated in the same way but comparing open field sessions from random cells whilst maintaining their temporal order (i.e., first arena vs second arena from a random cell) for each shuffle we did this 1000 times.

**Place cell criteria**. A cluster was classified as a place cell if it satisfied the following criteria in the session with the greatest number of spikes: (i) the peak to trough width of the waveform with the highest amplitude was >250 μs, (ii) the mean firing rate was >0.1 Hz but <10 Hz and (iii) the spatial information content was greater than 0.5 bits/second. Spatial information content was defined as:

$$\text{information content} = \sum P_i \left(\frac{R_i}{R}\right) \log_2 \left(\frac{R_i}{R}\right)$$

where $i$ is the voxel number, $P_i$ is the probability for occupancy of voxel $i$, $R_i$ is the mean firing rate for voxel $i$, and $R$ is the overall average firing rate[30]. In combination with these parameters we also manually refined the resulting place cell classification in order to resolve false positives and negatives.

**Place field characteristics**. A detailed description of the features used to describe place fields can be found in the Supplementary Methods sections: *Field detection*, *Field volume and density*, *Field orientation and size*, *Field elongation and sphericity*, *Field stability* and *Field distribution*. Briefly, for ease of comparison to existing literature most of the analyses we used were simply extensions of those used for two-dimensional data. Firing rate maps were constructed as above; these were thresholded at 20% of the peak firing rate and from this contiguous regions of high firing rate were then isolated. Regions larger than 64 voxels and visited more than five times were then analyzed as putative place fields. We calculated their volume (total volume of voxels in region), centroid (average voxel position), length along each Cartesian axis (side lengths of minimum enclosing cuboid) and diameter (diameter of minimum enclosing sphere).

In a departure from two-dimensional analyses we fitted a multivariate normal distribution (i.e., a 3D ellipse) to each field and from this extracted the field's principal axes (the three axes defining the ellipse), orientation (orientation of the field's longest axis), elongation (longest axis length divided by the mean of the other two, or second longest in the arena) and sphericity (ratio of surface area to a sphere of equivalent diameter). To determine whether fields were as or more spherical than would be expected by chance we used a shuffle procedure similar to

that reported previously[10] (Supplementary Methods: *Field elongation and sphericity*). To calculate fields per m³ for each maze we estimated the volume of the maze as the average volume of trajectories recorded in the maze (Supplementary Methods: *Field volume and density*). To determine whether fields were homogeneously distributed throughout the mazes we compared their distribution around the center of the maze to a 1000 shuffles of random points (shaded areas Fig. 5e, Supplementary Methods: *Field distribution*).

**Field orientation**. To generate the three-dimensional spherical heatmaps of field orientation and calculate the number of fields aligned with each maze axis we used the same analysis described for the behavioral data (see Methods: *Behavioral analyses* and Supplementary Methods: *Field orientation and size*). Briefly, we counted the number of fields that had an orientation within a 60° cone around each maze axis (i.e., a vertical field would have a vertically oriented longest axis which would also run parallel to the Z-axis). To compare between mazes we calculated a ratio of *XYZ* oriented fields to *ABC* oriented fields.

To determine whether fields were oriented parallel to one axis more than another we calculated 95% confidence intervals for each observed axis value using a bootstrapping procedure (error bars in Fig. 7c). If the value of one axis fell within the error bounds of another axis we considered them not significantly different. To determine whether fields were aligned with one or more axes at a frequency greater than would be expected by chance we randomly distributed 1000 points across the surface of a sphere 1000 times, and for each shuffle computed the number of points falling within the region corresponding to each axis. If the number of real fields aligned with a particular axis exceeded the 2.5th or 97.5th percentile of these shuffled counts (red areas in Fig. 7c) it was considered to be significantly under or overrepresented respectively. For each shuffle we also calculated the ratio of *XYZ* to *ABC* fields and if the observed ratio of a maze exceeded the 1st or 99th percentile of the ratios obtained in the shuffle (blue and red lines in Fig. 7c inset respectively) it was defined as significantly deviating from 1 (no axis bias of any kind) and indicated an overrepresentation of *ABC* or *XYZ* fields respectively.

**Spatial coding**. A detailed description and schematics of these methods can be found in Supplementary Methods: *Autocorrelation and spatial information*. Briefly, we generated three-dimensional autocorrelations using an extension of two-dimensional methods. We then extracted the central regions along each axis (i.e., a skewer running through the middle of the cube from one end to the other would represent one central section) to see whether correlations were higher along one axis and if this was related to the proximity of voxels (see Supplementary Fig. 20 for a schematic). We also projected data onto the three possible cardinal planes by averaging firing rate maps along each dimension (i.e., averaging 'floors' of the aligned lattice results in a single two-dimensional map in X and Y, which would be the average along the Z axis). After isolating fields (see Methods: *Place field characteristics*) we also summed the fields along each dimension, peak-normalized the resulting vectors and calculated the area under the curve (AUC) for each (Matlab function *trapz*).

**Field stability**. To test the stability of spatial representations within sessions we divided maze sessions into two halves of equal length (first 50% and second 50%) and computed the Pearson correlation (Matlab function *corr*) between the firing rate maps for these halves. As for *Recording stability*, these maps were generated by projecting data onto the Cartesian planes before calculating 2D firing rate maps in the standard manner; as bivariate histograms with 5 cm square bins smoothed using a Gaussian with 5 cm standard deviation (Matlab function *imgaussfilt*). Volumetric maps were also generated as multivariate histograms with 5 cm cubic bins smoothed using a Gaussian with 5 cm standard deviation (Matlab function *imgaussfilt3*). In both cases, spike and position data were truncated to include only the data falling within the lattice maze frame. In all cases, higher correlations were observed when dividing sessions based on odd and even minutes so these data are not shown. We compared these values to shuffled distributions generated by comparing session halves from random cells (i.e. first 50% vs second 50% from a random cell). For each projection and volumetric map we did this 1000 times with replacement.

**Reporting summary**. Further information on research design is available in the Nature Research Reporting Summary linked to this article.

## Data availability
A summary data set is available for download[32]. The full raw data set is available from the authors on request. The source data underlying Figs. 2b, 4c–f, 5b–d, 6a–d, 7b, c, 8b, c and Supplementary Figs. 1b–f, 4b, 5, 6b, 7b–d, 9c, e, f, 10a–e, 11a–j, 12b, 13a, b, d, e, 14c and 15b, c are provided as a Source Data file.

## Code availability
Matlab code is available for download which, in conjunction with the summary data set, can be used to regenerate all of the figures and analyses reported here[31]. Code to analyze raw data is available from the authors on request.

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

## Acknowledgements
The authors would like to thank Éléonore Duvelle for her continued advice throughout these experiments and Jim Donnett of Axona Ltd. for his expertise in designing and building the experimental equipment. This work was supported by a grant from Wellcome (103896AIA) to K.J.

## Author contributions
K.J. conceived the study and obtained funding, S.R., R.G., and K.J. designed the protocol, R.G., S.J.-A., K.M., A.L., and S.R. performed surgeries and recordings, R.G analyzed data. All authors interpreted data and discussed results. R.G. and K.J. wrote the manuscript. All authors commented and edited the manuscript.

## Competing interests
The authors declare the following competing interests: K.J. is a non-shareholding director of Axona Ltd.
