## [Peer Review File · Nature Communications]

Reviewers' Comments:

Reviewer #1:

Remarks to the Author:

In this paper, Grieves et al have recorded place cells from rats moving in lattice mazes covering a volume in space. They show that rats, in spite of being surface dwelling animals, are capable of volumetric representation of space but this representation is affected by constraints in movement that are introduced by the structure of the maze.

While the main claim of this paper is very important and well supported, there are major concerns (listed below) which must be addressed before the paper is accepted for publication.

The organization of the manuscript leaves the reader confused and frustrated. Details of experimental methods are arbitrarily distributed in results, figure legends, methods, supplementary methods and supplementary figure legends. Thus, in order to figure out how a specific thing was done, the reader is left to jump back and forth between the different sections. Clearly, this has also led to the authors being lost in the maze as well, since there are methods and figures detailed in one or more of these sections that are not discussed in the manuscript.

Major concerns-

Table S1 Shows that more than 150 cells each were recorded from 3 rats (852, 896, 926), over a large number of sessions (14, 5, and 9, respectively). The recordings use either 4 or 8 tetrodes. The authors are clearly aware that these recordings are likely to include same neurons sampled multiple times, as evidenced by the following statement:

"At the end of the recording session, the animals were removed from the apparatus and **the electrodes were lowered by at least 20 μm ** in order to maximise the chance of recording from a different population of cells on the following day. No attempt was made to track cells across days, **and thus a subset of the cells may have been recorded on more than one session**." (Supplement P75)

At 20 $\mu\text{m}/\text{day}$, 13 turnings between 14 sessions leads to a movement of 260 μm . The CA1 pyramidal cell layer is much narrower ($\sim 50 \mu\text{m}$). It is often possible to record pyramidal cells after moving so much, as the tetrode drags the layer down (evident in histology for rats 926, 896 and 853 shown in Fig S15).

This leads to two major problems: (1) inflation of degrees of freedom in statistics after classifying repeated recordings from the same neurons recorded on different days as different neurons and (2) data being overwhelmed by repeat samples from a single rat (e.g. 85% of the place cells for tilted maze come from a single rat).

The authors must show that neither of these two factors is leading them to the conclusions they report in the paper. An easy way of doing this is to include data from each tetrode from exactly one session (e.g. the session with most number of units/place cells) for the given maze in all population level statistics and figures (e.g. distribution of place fields in figure 2, shapes of place fields etc.). If the statistics remain significant despite this reduction in potentially repeated samples, then the reader can have confidence in the results not being coloured either by repeat sampling or domination of the dataset by a single rat.

Supp L163-168 "Next, the spikes emitted within the place field were randomly shuffled among the trajectories through this sphere (i.e. n random position data points within the field were selected as shuffled spike positions, where n = the number of spikes emitted within the field, Matlab function `randi`). We then recomputed the firing rate map for this shuffled field and extracted its elongation index as described above." This description sounds like the randomization is done uniformly within a sphere. We already know that the firing rates are not distributed uniformly in a place field (1D as well as 2D place fields can be approximated with gaussians; 3D place fields shown in the figures in the paper show similar lack of uniform distribution). Thus, the distribution used for this shuffle needs to be changed to match the underlying distribution as a function of distance from the field centre, while

keeping it symmetrical in the 3 axes. Otherwise, the randomization method will not create an appropriate control distribution to test whether spherical place fields could appear elongated due to sampling related issues.

L308-311 "Furthermore, we also observed the same sublinear relationship between environment and place field size as that study; place field size did not scale linearly with the environment but instead at a reduced rate, an effect which has also been reported previously in rats (22)."

The two scales of environments used here are a 1800mmX1800mm 2D arena and 970mm X 970 mm X 970mm 3D lattice. Firstly, two data points aren't sufficient to show sublinear relationship. If by "sublinear" the authors just mean that between the two sample points the scale factor is less than 1, the authors should rewrite to make this clear.

Secondly, the place field size isn't defined in the article. If place field volume is what the authors meant to talk about, they should mention so clearly and furthermore, define it for the place fields in the 2D open arena.

Lastly, how does one check if the scaling is linear or sublinear when going from a 2D to a 3D environment?

L342 "In the present experiment we found an increase in place field elongation in the vertical dimension, which was also represented less stably". The stability of representation mentioned here has not been discussed in the results of the article. Fig S11 does show results of analysis run for the same but no description of what shuffling procedure was used for the control distributions are presented in the legend. Authors are suggested to describe these results and the methods involved.

L56-61 "we found that place fields tended to be elongated along the axes of the maze (the directions aligned with the boundaries, and in which travel was easiest) with greater elongation for the vertical axis and a resultant lower spatial information and decoding accuracy. We then tilted the lattice so that the three planes of movement all had the same relationship to gravity, and were thus all equally easy (or hard) to traverse. We found that the elongation of the axes followed the tilt of the maze,". There is a possible confounding factor that can cause systematic errors leading to this conclusion. If the material used in the arena reflects LED light sufficiently (and most materials do, especially at close range) to cause enlargement of the "detected LED contour" selectively along the lattice axis, the arena shape itself will lead to errors in position estimation parallel to the major axes of the arena being larger than the errors in other directions. The authors need to verify if such errors can account for their claim of elongation of place fields along the arena axes.

Figure S2E shows that the rats run slowest in the tilted maze, and slower in 3D arenas than the 2D arena. The authors need to show that their main results remain unchanged after using only the comparable speed ranges in the three arenas.

L157-158 "It is unlikely these effects were due to inhomogeneous sampling (Fig. S9)" fig S9 takes care of inhomogeneous sampling of the movement along different axes (vertical vs horizontal) in aligned maze. It does not account for inhomogeneity of sampling at centre vs periphery or top vs bottom. The authors need to control for these.

Minor concerns-

There is no uniformity in the way the authors refer to areas and volumes. E.g. 16cm^3 (L474) looks like it stands for $16\text{cm} \times 16\text{cm} \times 16\text{cm}$ - otherwise, the rat would have to go in gaps less than 3cm to a side with its implant; in contrast, " 8000cm^3 " on L539 is " $20 \times 20 \times 20\text{cm}$ ". Throughout the manuscript, the authors should use the unambiguous notation used in the second example cited here to avoid confusion.

No firing rate maps for tilted maze have been shown. Authors should include a figure similar to S3 for titled maze's rate maps.

The manuscript has a number of grammatical/typographical errors e.g. L5 'We wirelessly recorded place cells in rats **at** they explored a cubic lattice climbing frame which could be aligned or tilted with respect to gravity', L402-402 "Except in the case of multivariate comparisons where we sought to determine any interaction effects, **then** we employed generalized linear models using SPSS."

Please consider proofreading the manuscript once more.

1000 mm² (also see the comment about confusing units used in describing areas and volumes) cues on walls of the room do not qualify as large as mentioned in supplementary methods L6, nor does this size sound correct from the photographs in Fig S1.

Fig 1C L89 says "frequency with which animals crossed between lattice layers in the two configurations" but there are three plots. Also, L91 says "ordered as in (B) and (C)" which implies that the first plot is for the open arena. Please correct the figure legend accordingly. Also, there is no mention of how layers along Z-axis were defined for open arena.

No mention of Grassberger-Procaccia algorithm is found in the cited paper (Ref. 11).

Furthermore, current description is not enough for a reader to understand the algorithm. Hence, Fig 2B and C as well as L124-127 of the main text have not been verified.

Similarly, there are other wrong citations. For example, L595 "method outlined by Zhang et al. (13)" Ref 13 is Wohlgemuth et al, and there is no Zhang et al paper in the reference list. L297-298 "method described by Zingg (10)" Ref 10 is Jovalekic et al., and Zingg is not in the reference list. There are other instances of inaccurate citations. Authors are advised to cross check all the references and citations.

No statistical tests were done to support the claim that 'in each case the median field centroids lay close to the maze center (Fig. S4F)'. Showing that the observed median values differ significantly from the distribution of median values calculated by shuffling the centroids of place fields inside the dwell time convex hull can be a way to support the claim made by authors.

For open arena, many quantities like volume of place field, diameter of the smallest sphere enclosing the field, layers along Z-axis, etc. have not been defined.

Axis ratio defined in methods (main text L580) is referred to as 'normalized ratio of XYZ to ABC' in Fig 2 legend (L145). Please stick to one terminology to avoid confusion.

FigS5D L504 the statistical test used in 'arena, aligned & tilted HDS: 0.057, 0.069 & 0.078, chance 99th percentile: 0.075' has not been defined in methods.

Supplement L531 for 'tilted correlation with predicted arena, aligned and tilted maps' the numbers mentioned are the ones for tilted (rotated) plot.

Fig S11 L652 'Values were higher in the arena than the lattice mazes' needs a one-sided test. No mention of whether this was done. Same is the case with L669 in supplementary methods Supplementary methods L145, the fraction in the exponent simplifies to $[-0.5(\{x:x<d\}/\sigma)]^2$ i.e. just the numerator.

Supplementary methods L244 "From this, we extracted the three voxel wide central portion along the X, Y and Z-axis, through the centre of the autocorrelogram....." and "We also extracted the central component parallel to each axis..." -the difference between these two wasn't very clear. A schematic diagram might aid the reader.

For analysis of coverage (Fig S2B L437), K-W test across sessions has been used. But no. of total sessions performed across all rats in tilted maze is 16. Non-parametric tests like K-W are not advisable for $N < 20$ (Zar, 2010). Hence, authors should avoid using these tests whenever the N s are small. If they still want to include such results, a clear statement, mentioning that the N s are small and hence the results should be taken into consideration with a pinch of salt, should be made.

L25-27 "This is important not just for spatial mapping per se but also because the spatial map may form the framework for other types of cognition in which information dimensionality is higher than in real space." takes things a bit too far in the speculative domain. You cannot easily extrapolate from representation of 3 dimensions of real space to multidimensional abstract space required for "other types of cognition".

L158-159 "Field heights were bimodal in the aligned lattice, suggesting that vertically elongated fields were longer than horizontally elongated ones" Technically inaccurate: bimodality indicates there are two groups, one with elongated and another without elongated field in vertical axis. It would be interesting to segregate the two groups and see if the group with elongated vertical axis was generally on the side of larger horizontal axes or distributed uniformly on horizontal projections.

L160-161 "Place field elongation in the lattice mazes was weakly but significantly positively correlated

with field centroid distance from maze center" Can this elongation be explained by uneven sampling near periphery/ lesser sampling in upper parts than lower parts?

L162 "In the square arena place fields were inhomogeneously distributed" this line is referring to place field orientations, and not locations as one might assume after reading.

The Bayesian decoding analysis was done on data from 1 rat each in aligned and tilted mazes (and 2 rats in 2D maze). The authors need to add in a caveat indicating that these results must be looked at cautiously as there is no way to know whether the results would hold true for multiple animals.

L397 "the Wilcoxon signed rank test (WSR, Matlab function ranksum)," Ranksum is not signed rank test; it is Wilcoxon rank sum test (equivalent to MannWhitney -U test). Matlab has a function called signrank. Did the authors mean that? L405 "DV explained by an IV" What is DV and IV?

The dimensions of a voxel are never explicitly mentioned, though they can be inferred to be 5cm x 5cm x 5cm from other descriptions.

Supp L23-25 "Hollow cubes were created by attaching red plastic tubes (length: 120mm, diameter: 10mm) using 6 or 4-way connectors. These cubes were then assembled into a 6x6x6 cubic maze (970x970x970mm)." $120 \times 6 = 720$; adding 10×6 for thickness of the orthogonal tubes takes it to 780. How did the numbers get to 970? Did the connectors account for 190mm? Instead of these details it might be easier to mention the actual dimensions of each cubical void in the lattice.

Supp L44-45 "wireless headstage (custom 64-channel, W-series, Triangle Biosystems Int., Durham, NC)" What were the gain and filter settings on the headstage?

Supp L375-376 "). This was computed for 500 logarithmically spaced points between 0-500 Hz." But the data were resampled at 250Hz, according to L351-352.

As mentioned in the general comments, methods and results are distributed throughout the manuscript non-intuitively. For e.g- the methods involved in field elongation and sphericity analysis have been split into main text under the heading of "Place field features" and supplementary methods under the heading of "Field sphericity" without any reference to each other. Even the headings aren't consistent enough to form a link.

This confusing layout has also led to confusion of authors and some figures and methods are not being referred to in results and discussion. For e.g. Zingg's shape classification has been described and a figure (S10) has been included but the results are limited only to figure legend. This analysis has not been referred to or put into context throughout the manuscript. Similarly, Fig. S11 has not been referred to anywhere in the manuscript. As mentioned in one of the earlier comments, the results from the same are being used in discussion but there is no reference made to the figure.

Authors are requested to rearrange the manuscript to ensure better flow of reading and refer to all sections that are related to one other, the way they have done for sections like "Apparatus".

References-

Zar JH (2010) Biostatistical Analysis. fifth ed., Prentice-Hall.

Reviewer #2:

Remarks to the Author:

In this manuscript Grieves et al explores the representation of three-dimensional (3D) space by place cells in rats using a novel 3D maze. Their findings confirm previous observation in bats, which showed that place cells can encode 3D space volumetrically. The rest of the paper focuses on whether place cells in 3D are isotropic (i.e. have similar size in all dimensions). The authors show that place cells are elongated along the vertical dimension if the movement of the rat along this dimension is more difficult. In contrast, if the movement is equally difficult in all 3-dimensions – the encoding appears to be isotropic – similarly to what was found in bats. This observation was already made in two previous papers from the Jeffery lab. Specifically, Casali et al and Hayman et al, showed that vertical elongation of place fields on a vertical plane depends on how difficult the vertical movement is (when movement

was easier – place fields were no longer elongated). Therefore, the current paper adds very little to this conclusion. Furthermore, the entire results section that attempts to quantify whether the encoding is isotropic is very hard to follow and is full of inconsistencies, as I describe below. For instance sometimes, the authors claim that “place fields were elongated in all dimensions”, but at the same time say that this occurs mostly in vertical dimension.

Major comments

What are the distributions of firing rates of place cells in different recordings apparatuses? What is the stability of place cells under the different recording conditions?

All the analyses related to the 2D arena seem to be unnecessary and might lead to misleading conclusions. For instance, there is a whole paragraph (Lines 162-169) making the point that in a 2D arena there are more place fields along X,Y axis compared to the Z axis. This result is completely inconclusive because there is practically no behavioral coverage in the Z axis in a 2D arena. Isn't it the very reason the authors recorded in a volumetric maze? What is the relevance of the A,B,C axes for the arena? I suggest removing entirely all the analysis related to 2D from the paper. The only reason to include the 2D data would be to compare the statistics of *individual* place cells recorded in both 2D and maze configuration (e.g. is there correlation in place fields size, elongation etc, across different mazes?).

The analyses related to place-field orientation along different dimensions are potentially interesting, but require much more explanation in the main text, and in the figure legend. It would be useful to show some schematics of how one gets from a traditional firing rate-map to the spheres shown in Fig.2D. This is not a common analysis in the field, and I don't think that the explanations in Line 561 of the Methods are clear enough. For instance – what is the definition of “place-field eigenvectors” (Methods, Line 563?). Also, the description of the behavioral analyses (Methods, 480-488) and the spherical representation in Fig1D are not clear.

The results section of the paper often reads like a list of statistical tests, and it is often not clear what should the reader conclude from the different comparisons. Furthermore, many of the relevant analysis are scattered across supplementary figures. It took me almost the entire paper to get to the most important figure – Fig3A. It shows that place-fields are larger in Z (than in X,Y) for aligned but not tilted maze. This result should be highlighted properly, and in my view come before Fig. 2D,E. Analyses of autocorrelation, information content etc take an entire page of the main text while being redundant with the simple point made in Fig3A, and should go to the supplementary.

Line 145 – the results of this section contradicts its caption (“Place fields were elongated in all dimensions”) and suggest that place fields are in fact elongated only in vertical dimension and not in all dimensions (Line 159, FigS5B). So are place fields elongated in other dimension (except vertical?) In the discussion the authors say again (Line 315): “Elongation did not occur in every direction but was almost always in the direction of the maze/axes boundaries”. I couldn't find any analyses to back this claim that there is elongation in some other directions, and not in vertical dimension only. Further, elongation in the vertical dimension seem to contradict the claim in the discussion that elongation occurs in the direction of the travel, because vertical dimension is least traveled (in the aligned maze).

Fig. S5 B,C – suggest that the fields are elongated along some dimension. Line 155: “With elongated indices that significantly deviated from 1”. Deviations from 1 could result from noisy estimate of the

place-field shape, due to firing-rate variability or inhomogeneous behavioral coverage. The author should compare the data to elongation index of simulated spherical fields. This should be done by taking the distribution of place-field firing rates from the data + noise (e.g. Poisson noise) and superimposing the spherical fields on actual behavioral trajectories. The authors should also show a histogram of the elongation indices, relative to a control distribution.

Same should be done for sphericity index (Fig S5C). Can down-sampling of horizontal movements explain the elongation along the Z dimension (Fig. S9)?

It would be nice to see examples of elongated place fields (not as convex hulls – which are very sensitive to outliers, but as raw data with behavioral trajectory + spikes as well as with firing-rate maps in different projections).

I also suggest to make Fig S4, and S5 main figures. In S4, it would be useful to show histograms for data presented in panels B-E, instead of bar-graphs (for the aligned versus tilted maze cases).

From the histogram of field extents in Fig. S5D left – it seems that the distribution of fields extents in Z is bimodal – most of the fields are not elongated (compared to X,Y), but a small proportion are. What is special about these place fields? Do place fields become more elongated in Z as a function of their height in the maze? Analysis shown in Fig.S5F should be conducted separately for place-field extent in each dimension versus distance along this dimension.

For cells with multiple fields - is there a correlation between size of multiple fields, or their elongation along a specific dimension? Do the authors have the same cells recorded in both tilted and aligned mazes? If so, do cells with elongation in Z in the aligned maze show elongation in the tilted maze?

Line 229 . The authors show that spatial and mutual information is higher for the Z dimension, but paradoxically conclude that this means that Z is encoded more poorly. This is a very problematic claim. Larger place-fields of individual cells do not necessarily mean degraded population coding - it depends on factors like the distribution of firing-rates and density of place-fields (see Kim, Ganguli, Frank Journal of Neuroscience 2012).

Fig3E bottom shows that the decoding error is the same for all three axes in all configurations, but the quality factor is different. Why is that? Does the behavioral coverage affect the prior? Would the conclusion be different if the behavior is matched as in Fig S9?

Minor comments

Line25: "This is important not just for spatial mapping per se but also because the spatial map may form the framework for other types of cognition in which information dimensionality is higher than in real space." Not clear why this is relevant.

Line46: Implies a difference between the finding in rats and bats. It is not clear what this difference is, since as the authors mention in a few sentences earlier – Casali et al (ref 7) demonstrated that rat place cells can be isotropic similarly to bat place cells.

Fig S1 left, middle should be part of the main figure.

Fig S2 panel F: is left – aligned, right – tilted? Not clear from the legends.

Fig1, and Fig S2: It's not clear from the schematics what do the rotated axes A,B,C correspond to. Perhaps using different colors/transparent cubes would help.

What is Fig1C? Is it a 2D arena? It's not clear at all that you have a 2D arena and a 3D maze. Showing 3D coverage data for a 2D arena (e.g. "layer" crossing Fig1 C left) is not informative and confusing.

Fig1A – what is the colorbar in column 1? Looks like blue = many spikes. Red=few spikes. It's confusing given that in column 2 the colorbar seems to be the opposite.
What are min/max firing-rates of cells in A column 4?

Fig. S3 – the two sessions from arena recordings appear as if they correspond to different heights of the maze.

Line 383 – spatial coding during volumetric navigation in bats does not seem to be conditioned on environmental geometry, whereas the current paper (as well as two previous papers from the same group) shows that in rats the conclusion of any volumetric study strongly depends on the exact geometry of the environment. Therefore, it is unclear how this "undermines" the necessity to study spatial representation in animals such as bats, whose movement in 3D is unconstrained.

Line 388-387 The logic of this sentence is unclear.

Reviewer #3:

Remarks to the Author:

Grieves et.al. perform the first ever recordings of the rat hippocampus during volumetric navigation, using telemetry and a 3D climbing lattice. They find 3D place cells in this environment that span the entire space, much like the types previously found in flying bats. Unlike in bats, they find non-isometric distortions of these fields, with the z-axis given special "treatment" by the hippocampus. This is an interesting result that serves as a nice link between previous work in rats and bats. It helps one form a single, coherent model of hippocampal representations across species. My comments are related to issues with data presentation.

1) My major issue with the paper is that it's a bit hard to read. Most of the Results section goes through a wide array of statistics, which all compare three arenas to one another. These are all presented parenthetically in the text in a way that makes the logic hard to follow after a while. My strong suggestion is to make a table of all the statistics that contains three columns for the three environments, and to remove these statistics from the text itself. The text can then be much more concise in summarizing the main results.

2) Partly related to this, Figure 1 presents data in a somewhat confusing and redundant way. I realize that layer crossings and proportion of total time are different measures, but they are probably strongly correlated to one another. It's not clear why both are presented, and why A,B,C are omitted from Fig 1B but not 1E.

3) The authors keep referring to gravity throughout the paper, and keep switching between referencing gravity and ease of motion. The issue is that the effects of gravity are not demonstrated at all, and the observed effects are likely due to all the movement non-uniformities. There is evidence in the literature for field elongations along common trajectories in other types of environments. But talking about gravity, the authors some sort of a low-level vestibular effect of place cells. I think that it's fine to mention gravity as a possible contribution in the discussion, but that the paper shouldn't be mentioning gravity so casually throughout the text.

4) The authors report that autocorrelation values are different between the three slanted axes (lines 194-196). This suggests that a) not enough data was collected to balance our individual rats'

idiosyncrasies and/or b) there is some asymmetry in the arena itself that's not accounted for. This asymmetry is not addressed, which is a problem because the result goes against the main logic of the paper (i.e. how can we believe that the Z axis statistics are different due to the vertical alignment of this axis, as opposed to other unknown factors)?

5) In Fig 3E, the quality factor is lower for the Z-axis than for the X or Y axes. This is consistent with the main results of the paper. However, the data come from a total of 4 sessions from a single rat and the effect seems to be driven entirely by a single outlier. It's a bit hard to trust this result. On a related note, are mean or median statistics being used here to compare the axes?

Response to reviewers

Reviewer #1:

1. *The organization of the manuscript leaves the reader confused and frustrated. Details of experimental methods are arbitrarily distributed in results, figure legends, methods, supplementary methods and supplementary figure legends. Thus, in order to figure out how a specific thing was done, the reader is left to jump back and forth between the different sections. Clearly, this has also led to the authors being lost in the maze as well, since there are methods and figures detailed in one or more of these sections that are not discussed in the manuscript.*

We have tried our best to simplify the manuscript and make it clear which sections refer to each other. In main methods and results (where applicable) we give the section heading in supplementary data that the reader should refer to. Statistics have been removed from figure legends as far as possible, we have placed the majority of main text statistical results in Table 1 (as per Reviewer 3's suggestion) and we have removed repetitive or redundant results. We have included in the main methods a description of every analysis reported in the main text and figures: in some cases these are briefer versions of more detailed descriptions in the supplementary methods. We have checked over the supplementary data and removed sections no longer related to the manuscript. Some sections in supplementary data, such as the sections on theta phase and running speed relationships, we did not feel were central to the paper but may be of interest to specific readers (e.g., for future computational modelling). We have tried to refer to these in the main text where relevant but they are not discussed in detail there.

2. *Table S1 Shows that more than 150 cells each were recorded from 3 rats (852, 896, 926), over a large number of sessions (14, 5, and 9, respectively). The recordings use either 4 or 8 tetrodes. The authors are clearly aware that these recordings are likely to include same neurons sampled multiple times, as evidenced by the following statement:*

*"At the end of the recording session, the animals were removed from the apparatus and ****the electrodes were lowered by at least 20 μm **** in order to maximise the chance of recording from a different population of cells on the following day. No attempt was made to track cells across days, ****and thus a subset of the cells may have been recorded on more than one session****. Rats were tested until cells were no longer observed (median, min & max: 3, 1 & 14 sessions)."*
(Supplement P75)

At 20 $\mu\text{m}/\text{day}$, 13 turnings between 14 sessions leads to a movement of 260 μm . The CA1 pyramidal cell layer is much narrower ($\sim 50 \mu\text{m}$). It is often possible to record pyramidal cells after moving so much, as the tetrode drags the layer down (evident in histology for rats 926, 896 and 853 shown in Fig S15).

This leads to two major problems: (1) inflation of degrees of freedom in statistics after classifying repeated recordings from the same neurons recorded on different days as different neurons and

(2) data being overwhelmed by repeat samples from a single rat (e.g. 85% of the place cells for tilted maze come from a single rat).

The authors must show that neither of these two factors is leading them to the conclusions they report in the paper. An easy way of doing this is to include data from each tetrode from exactly one session (e.g. the session with most number of units/place cells) for the given maze in all population level statistics and figures (e.g. distribution of place fields in figure 2, shapes of place fields etc.). If the statistics remain significant despite this reduction in potentially repeated samples, then the reader can have confidence in the results not being coloured either by repeat sampling or domination of the dataset by a single rat.

We looked carefully at the cluster spaces and are confident that cells were not recorded on more than one day, but to be sure, we have also analysed data from only one session per tetrode per rat as suggested. We find the results essentially unchanged. We have added a mention of this to the text:

“Similar results to those reported in main text were also observed when only analyzing one session per animal (the session with the most place cells). These analyses can be replicated using the provided data set and code.”

We have also supplied the actual analyses for the reviewers to evaluate (figures available at the end of this document). We felt including these in the paper itself would add unnecessarily to its length.

3. *Supp L163-168 “Next, the spikes emitted within the place field were randomly shuffled among the trajectories through this sphere (i.e. n random position data points within the field were selected as shuffled spike positions, where n = the number of spikes emitted within the field, Matlab function `randi`). We then recomputed the firing rate map for this shuffled field and extracted its elongation index as described above.”*

This description sounds like the randomization is done uniformly within a sphere. We already know that the firing rates are not distributed uniformly in a place field (1D as well as 2D place fields can be approximated with Gaussians; 3D place fields shown in the figures in the paper show similar lack of uniform distribution). Thus, the distribution used for this shuffle needs to be changed to match the underlying distribution as a function of distance from the field centre, while keeping it symmetrical in the 3 axes. Otherwise, the randomization method will not create an appropriate control distribution to test whether spherical place fields could appear elongated due to sampling related issues.

At the reviewer’s suggestion we have replaced this analysis with one where spikes are distributed using a multivariate Gaussian process, which should approximate the internal dynamics of the place fields. This greatly increases the proportion of cells which are more spherical than chance (from ~5% to ~40%), but these are still much smaller proportions than observed in bats (~95%).

4. L308-311 *“Furthermore, we also observed the same sublinear relationship between environment and place field size as that study; place field size did not scale linearly with the environment but instead at a reduced rate, an effect which has also been reported previously in rats (22).”*

The two scales of environments used here are a 1800mmX1800mm 2D arena and 970mm X 970 mm X 970mm 3D lattice. Firstly, two data points aren't sufficient to show sublinear relationship. If by “sublinear” the authors just mean that between the two sample points the scale factor is less than 1, the authors should rewrite to make this clear.

We have removed “sublinear” and now say that the scale factor is less than 1, as suggested.

Secondly, the place field size isn't defined in the article. If place field volume is what the authors meant to talk about, they should mention so clearly and furthermore, define it for the place fields in the 2D open arena.

We have added the following sentence to make this clearer and to clarify what we mean by size:

“If we treat our arena as a short volume instead of a 2D plane the predicted volume of fields in the lattice would be much larger than those observed.”

*This was the only section of text referring to place field ‘size’. Place field volume is defined in Methods: *Place field characteristics*.*

5. *Lastly, how does one check if the scaling is linear or sublinear when going from a 2D to a 3D environment?*

*We estimated the practical volume of each maze based on animals' trajectories (i.e. average convex hull volume of session trajectories) this is now described in Methods and Supp. Methods: *Field volume and density*. Using these values we computed place fields per m³ which we used to compare across mazes. By sublinear we mean that the volume scaling of fields is less than 1 between mazes, which has now been clarified (see previous comment).*

6. L342 *“In the present experiment we found an increase in place field elongation in the vertical dimension, which was also represented less stably”. The stability of representation mentioned here has not been discussed in the results of the article. Fig S11 does show results of analysis run for the same but no description of what shuffling procedure was used for the control distributions are presented in the legend. Authors are suggested to describe these results and the methods involved.*

*This analysis has now been replaced with a more sensitive and detailed one. These new shuffling and analysis methods have been added to methods sections entitled ‘*Field stability*’ and ‘*Recording stability*.’ The results of these analyses are reported more centrally at the end of the Result section: *Spatial coding was less accurate along the vertical dimension*.*

7. L56-61 *“we found that place fields tended to be elongated along the axes of the maze (the directions aligned with the boundaries, and in which travel was easiest) with greater elongation for the vertical axis and a resultant lower spatial information and decoding accuracy. We then*

tilted the lattice so that the three planes of movement all had the same relationship to gravity, and were thus all equally easy (or hard) to traverse. We found that the elongation of the axes followed the tilt of the maze,”

There is a possible confounding factor that can cause systematic errors leading to this conclusion. If the material used in the arena reflects LED light sufficiently (and most materials do, especially at close range) to cause enlargement of the “detected LED contour” selectively along the lattice axis, the arena shape itself will lead to errors in position estimation parallel to the major axes of the arena being larger than the errors in other directions. The authors need to verify if such errors can account for their claim of elongation of place fields along the arena axes.

We did not observe reflections during recording, because the experimental apparatus and room were designed specifically to optimise tracking. The maze material is not particularly reflective, and in any case the effect of reflections would be very small as the animal’s position is tracked using the centroid of 4 LEDs mounted on the headstage, so a reflection would have to be very bright and large in surface area to drag that centroid off-center. If so it would have been noticed by the experimenter: also, it would have equally affected both behavioural tracking (Fig. 1) and spiking activity (Fig. 7), but this is not the case in the aligned lattice. For all these reasons we consider this to be a very unlikely confound. Fig. S16 details some tests to demonstrate the lack of reflections and tracking artefacts: long sections of video footage from our experimental setup are also shown in Supplementary Video 2, around time mark 2:40 there are some examples of reflections on the arena walls which are not tracked by the system, and additional examples can be seen throughout the video.

8. *Figure S2E shows that the rats run slowest in the tilted maze, and slower in 3D arenas than the 2D arena. The authors need to show that their main results remain unchanged after using only the comparable speed ranges in the three arenas.*

Although animals move differently along specific axes (i.e. faster along X and Y than Z in the aligned lattice) overall speed profiles do not differ between the three environments. We have made this clearer at the start of results and added this information to Fig. S1 and Supp. Data: *movement patterns in the lattice mazes:*

“Animals spent significantly less time moving at slower speeds ($F(24,1225) = 126.0, p < .0001, \eta_p^2 = 0.71$) but this did not differ significantly between environments ($F(2,1225) = 0.2, p = .82, \eta_p^2 < 0.001$) nor was there a significant interaction between the two ($F(48,1225) = 1.2, p = .17, \eta_p^2 = 0.045$, Univariate ANOVA comparing effects of speed and environment on dwell time, Fig. S1f).”

9. *L157-158 “It is unlikely these effects were due to inhomogeneous sampling (Fig. S9)” fig S9 takes care of inhomogeneous sampling of the movement along different axes (vertical vs horizontal) in aligned maze. It does not account for inhomogeneity of sampling at centre vs periphery or top vs bottom. The authors need to control for these.*

The data have now also been downsampled to match movements in the inner and outer 50% of the lattice, and the top and bottom 50% of the lattice. These results can be seen in supplementary Figure S6. No change was found using any of the downsampling approaches.

10. *There is no uniformity in the way the authors refer to areas and volumes. E.g. 16cm^3 (L474) looks like it stands for $16\text{cm} \times 16\text{cm} \times 16\text{cm}$ - otherwise, the rat would have to go in gaps less than 3cm to a side with its implant; in contrast, " 8000cm^3 " on L539 is " $20 \times 20 \times 20\text{cm}$ ". Throughout the manuscript, the authors should use the unambiguous notation used in the second example cited here to avoid confusion.*

This has been corrected in favour of the unambiguous second style (i.e. $100 \times 100 \times 100\text{cm}$).

11. *No firing rate maps for tilted maze have been shown. Authors should include a figure similar to S3 for titled maze's rate maps.*

These plots can now be found in Fig. 3, Fig. S3 and Supplementary Video 4.

12. *The manuscript has a number of grammatical/typographical errors e.g. L5 'We wirelessly recorded place cells in rats **at** they explored a cubic lattice climbing frame which could be aligned or tilted with respect to gravity',*

L402-402 *"Except in the case of multivariate comparisons where we sought to determine any interaction effects, **then** we employed generalized linear models using SPSS."* Please consider proofreading the manuscript once more.

These errors have been corrected and the manuscript has been thoroughly proofread.

13. *1000mm^2 (also see the comment about confusing units used in describing areas and volumes) cues on walls of the room do not qualify as large as mentioned in supplementary methods L6, nor does this size sound correct from the photographs in Fig S1.*

These unit errors have been corrected in Supplementary Methods: Apparatus.

14. *Fig 1C L89 says "frequency with which animals crossed between lattice layers in the two configurations" but there are three plots. Also, L91 says "ordered as in (B) and (C)" which implies that the first plot is for the open arena. Please correct the figure legend accordingly. Also, there is no mention of how layers along Z-axis were defined for open arena.*

The layer analysis has been removed because it essentially repeats the dwell time analysis and because this was overly confusing. Fig. 2 has been reworked and the figure legend has been clarified.

15. *No mention of Grassberger-Procaccia algorithm is found in the cited paper (Ref. 11). Furthermore, current description is not enough for a reader to understand the algorithm. Hence, Fig 2B and C as well as L124-127 of the main text have not been verified.*

The reference section for the supplementary file was missing but has been restored now. However, for clarity and ease of understanding we have replaced the GP algorithm with a

simpler shuffle based approach which indicates fields are distributed around the centre of each maze axis; Fig. 5c-d, Supp. Methods: *Field distribution*.

16. Similarly, there are other wrong citations. For example, L595 “method outlined by Zhang et al. (13)” Ref 13 is Wohlgemuth et al, and there is no Zhang et al paper in the reference list.

The correct reference section is included in the supplementary file now.

17. L297-298 “method described by Zingg (10)” Ref 10 is Jovalekic et al., and Zingg is not in the reference list. There are other instances of inaccurate citations. Authors are advised to cross check all the references and citations.

The correct reference section is included in the supplementary file now.

18. No statistical tests were done to support the claim that ‘in each case the median field centroids lay close to the maze center (Fig. S4F)’. Showing that the observed median values differ significantly from the distribution of median values calculated by shuffling the centroids of place fields inside the dwell time convex hull can be a way to support the claim made by authors.

A shuffle based approach has now been added which indicates fields are distributed around the centre of each maze axis; Fig. 5c-d, Supp. Methods: *Field distribution*.

19. For open arena, many quantities like volume of place field, diameter of the smallest sphere enclosing the field, layers along Z-axis, etc. have not been defined.

As above, layer analyses for the arena have been removed. Quantities calculated for the arena were calculated as described in the text, no exception was made for the arena unless otherwise stated (i.e. when calculating field elongation we only used the 2 longest axes in the arena but all 3 in the lattice mazes). For clarification, the following text has been added to Supp. Methods: *Field detection*:

“Although the open-field arena was flat, animals were free to move vertically in the area (i.e. crouching, standing and rearing) and so we treated the arena data as a thin volume; nevertheless, the analyses for the z dimension (height) must be treated with caution.”

20. Axis ratio defined in methods (main text L580) is referred to as ‘normalized ratio of XYZ to ABC’ in Fig 2 legend (L145). Please stick to one terminology to avoid confusion.

Axis ratio has been simplified to XYZ/ABC: the definition has been updated in both locations and a plot has been added to Fig. 2b and Fig. 7c for a graphical illustration.

21. FigS5D L504 the statistical test used in ‘arena, aligned & tilted HDS: 0.057, 0.069 & 0.078, chance 99th percentile: 0.075’ has not been defined in methods.

This was the “Hartigan dip statistic” which tested whether the Z distribution was more bimodal than X or Y. However we have replaced it with a more sensitive bootstrapped modality procedure which is described in Supp. Methods: *Field elongation*.

22. Supplement L531 for 'tilted correlation with predicted arena, aligned and tilted maps' the numbers mentioned are the ones for tilted (rotated) plot.

This error has been corrected and the true tilted values have been added.

23. Fig S11 L652 'Values were higher in the arena than the lattice mazes' needs a one-sided test. No mention of whether this was done. Same is the case with L669 in supplementary methods

This analysis has been re-run using a slightly different approach to gain better detail of the effects. Recording stability (i.e. comparing arena 1 to arena 2) and field stability (i.e. comparing first half of lattice to second half) have been separated in supplementary methods and data for clarity. These have also been expanded to provide more information. Taking the reviewer's comment into account, we have tested the distributions shown with a one-sided Wilcoxon rank sum test in addition to measuring the 95th percentile of a shuffle distribution (Fig. S4 and Fig. S11). We hope this text and analysis are more understandable.

24. Supplementary methods L145, the fraction in the exponent simplifies to $\left[\frac{-0.5(x-d)/\sigma}{\sigma} \right]^2$ i.e. just the numerator.

This has been updated.

25. Supplementary methods L244 "From this, we extracted the three voxel wide central portion along the X, Y and Z-axis, through the centre of the autocorrelogram....." and "We also extracted the central component parallel to each axis..." -the difference between these two wasn't very clear. A schematic diagram might aid the reader.

An explanatory figure (Fig. S19) has been added to supplementary data and the text has been expanded to try and clarify the process.

26. For analysis of coverage (Fig S2B L437), K-W test across sessions has been used. But no. of total sessions performed across all rats in tilted maze is 16. Non-parametric tests like K-W are not advisable for $N < 20$ (Zar, 2010). Hence, authors should avoid using these tests whenever the N s are small. If they still want to include such results, a clear statement, mentioning that the N s are small and hence the results should be taken into consideration with a pinch of salt, should be made.

We have added a caution in each case, such as:

"...although the small sample sizes ($N = 34$ & 16 sessions respectively) should be taken into consideration when interpreting this result."

These two examples appear to be the only non-parametric tests involving small sample sizes that are not permutation tests.

27. L25-27 "This is important not just for spatial mapping per se but also because the spatial map may form the framework for other types of cognition in which information dimensionality is higher than in real space." takes things a bit too far in the speculative domain. You cannot easily

extrapolate from representation of 3 dimensions of real space to multidimensional abstract space required for "other types of cognition".

We have clarified by amending as follows:

"The question arises as to whether this map is three-dimensional, and if so whether the properties are the same in all dimensions, and how information is integrated across these dimensions".

We agree that such extrapolation may not be possible, but it remains a valid hypothesis.

28. L158-159 *"Field heights were bimodal in the aligned lattice, suggesting that vertically elongated fields were longer than horizontally elongated ones"* Technically inaccurate: bimodality indicates there are two groups, one with elongated and another without elongated field in vertical axis. It would be interesting to segregate the two groups and see if the group with elongated vertical axis was generally on the side of larger horizontal axes or distributed uniformly on horizontal projections.

Due to the weak nature of the bimodality we do not feel this effect is strong enough to pursue further. The figure is still included in Fig. S7a as it could be of interest to some readers. However, it is worth noting that the rightmost peak corresponds approximately to the vertical extent of the lattice and so is likely due to an edge effect of the apparatus clipping fields that we know are generally longer in the vertical dimension.

29. L160-161 *"Place field elongation in the lattice mazes was weakly but significantly positively correlated with field centroid distance from maze center"* Can this elongation be explained by uneven sampling near periphery/ lesser sampling in upper parts than lower parts?

As also explained above, the data have now also been downsampled to match movements in the inner and outer 50% of the lattice, and the top and bottom 50% of the lattice. These results can be seen in supplementary Fig. S6. No change was found using any of the downsampling approaches.

30. L162 *"In the square arena place fields were inhomogeneously distributed"* this line is referring to place field orientations, and not locations as one might assume after reading.

This has been clarified to:

"In the square arena the three dimensional orientations of place fields were not random; instead the majority of fields had their longest axis running parallel to the X and Y axes..."

31. *The Bayesian decoding analysis was done on data from 1 rat each in aligned and tilted mazes (and 2 rats in 2D maze). The authors need to add in a caveat indicating that these results must be looked at cautiously as there is no way to know whether the results would hold true for multiple animals.*

We tried to extend this analysis to include more sessions/animals in a variety of ways but this has not been successful. Given the very low numbers of animals in each group we have removed this analysis completely.

32. L397 “the Wilcoxon signed rank test (WSR, Matlab function ranksum),” Ranksum is not signed rank test; it is Wilcoxon rank sum test (equivalent to MannWhitney -U test). Matlab has a function called signrank. Did the authors mean that?

These tests were confused, this has been corrected in Table 2 (Methods).

33. L405 “DV explained by an IV” What is DV and IV?

These have been expanded to dependent variable and independent variable respectively.

34. The dimensions of a voxel are never explicitly mentioned, though they can be inferred to be 5cm x 5cm x 5cm from other descriptions.

This has been added to Methods: *Firing rate maps*:

“The firing rate in each voxel (50×50×50mm) was calculated as...”

And to Supp. Methods: *Field detection*:

“...we looked for areas of more than 64 contiguous voxels (voxels were 50×50×50mm) with a firing rate...”

35. Supp L23-25 “Hollow cubes were created by attaching red plastic tubes (length: 120mm, diameter: 10mm) using 6 or 4-way connectors. These cubes were then assembled into a 6×6×6 cubic maze (970×970×970mm).” 120×6 = 720; adding 10 x 6 for thickness of the orthogonal tubes takes it to 780. How did the numbers get to 970? Did the connectors account for 190mm? Instead of these details it might be easier to mention the actual dimensions of each cubical void in the lattice.

The 120mm length was incorrect and should have been 150mm: each connector adds a spacing of 10mm with 7 connectors required along one side of the lattice. Together this gives 150*6 + 10*7 = 970mm. These measurements have been corrected/clarified in Supp. Methods: *Apparatus*.

36. Supp L44-45 “wireless headstage (custom 64-channel, W-series, Triangle Biosystems Int., Durham, NC)” What were the gain and filter settings on the headstage?

This information has been added to Supp. Methods: *Recording setup and procedure*:

“Unfiltered signals were sampled at 50kHz, amplified 100 times and transmitted...”

37. Supp L375-376 “). This was computed for 500 logarithmically spaced points between 0-500 Hz.” But the data were resampled at 250Hz, according to L351-352.

This has been corrected to:

“This was computed for 500 logarithmically spaced points between 0-250 Hz.”

38. As mentioned in the general comments, methods and results are distributed throughout the manuscript non-intuitively. For e.g- the methods involved in field elongation and sphericity analysis have been split into main text under the heading of “Place field features” and

supplementary methods under the heading of “Field sphericity” without any reference to each other. Even the headings aren’t consistent enough to form a link.

We have tried to make the results and methods easier to follow. Section headings are provided in text when referring to Supplementary Methods or Data. The main text methods should now contain a description of all the methods necessary to understand the main text, although in some cases we provide a more detailed description in Supplementary methods. Section headings should be more informative about their contents and where possible the headings match between sections (i.e. “*Supp. Methods: Field elongation and sphericity*” and “*Supp. Data: Field elongation*” both support the main text section “*Place fields were elongated rather than spherical*”)

39. *This confusing layout has also led to confusion of authors and some figures and methods are not being referred to in results and discussion. For e.g. Zingg’s shape classification has been described and a figure (S10) has been included but the results are limited only to figure legend. This analysis has not been referred to or put into context throughout the manuscript.*

The Zingg shape analysis and sphericity analysis were mistakenly cut from the main text but have been added back into the results section “*Place fields were elongated rather than spherical*”

40. *Similarly, Fig. S11 has not been referred to anywhere in the manuscript. As mentioned in one of the earlier comments, the results from the same are being used in discussion but there is no reference made to the figure.*

Both recording stability (i.e. between arena 1 and arena 2) and field stability (i.e. between session first half and second half) have been expanded with a more detailed analysis. These are both now discussed in the main results section and have their methods described in main methods with appropriate section headings.

41. *Authors are requested to rearrange the manuscript to ensure better flow of reading and refer to all sections that are related to one other, the way they have done for sections like “Apparatus”.*

As described above, we have tried to make the section headings as helpful as possible and we use these throughout to direct readers to appropriate sections elsewhere.

References-

Zar JH (2010) Biostatistical Analysis. fifth ed., Prentice-Hall

Reviewer #2

1. *In this manuscript Grieves et al explores the representation of three-dimensional (3D) space by place cells in rats using a novel 3D maze. Their findings confirm previous observation in bats, which showed that place cells can encode 3D space volumetrically. The rest of the paper focuses on whether place cells in 3D are isotropic (i.e. have similar size in all dimensions). The authors show that place cells are elongated along the vertical dimension if the movement of the rat along this dimension is more difficult. In contrast, if the movement is equally difficult in all 3-dimensions – the encoding appears to be isotropic – similarly to what was found in bats. This observation was already made in two previous papers from the Jeffery lab.*

Specifically, Casali et al and Hayman et al, showed that vertical elongation of place fields on a vertical plane depends on how difficult the vertical movement is (when movement was easier – place fields were no longer elongated). Therefore, the current paper adds very little to this conclusion.

Our results are similar to those made in bats but since they are a different species with a very different movement ecology – specifically, bats fly through volumetric space whereas rats, like humans, are surface-travelling – we feel that this is an important finding rather than simply a confirmation.

The other experiments listed by the reviewer were all conducted in two-dimensional, planar environments and the reviewer's conclusion is not quite correct – place fields were elongated on the pegboard (Hayman et al) but normal size and shape, and yet sparser, on the climbing wall (Casali et al) – despite both apparatuses being difficult to explore. The prediction for volumetric space is thus not clear, and needed to be tested experimentally (and one could argue that even if it were clear it should nevertheless be tested). Our results point to rats forming a different representation for volumetric space that reflects its structural characteristics. Taken together the results for all three experiments point to a role for environment structure in the nature of place field encoding – we think this is an important conclusion.

2. *Furthermore, the entire results section that attempts to quantify whether the encoding is isotropic is very hard to follow and is full of inconsistencies, as I describe below. For instance sometimes, the authors claim that “place fields were elongated in all dimensions”, but at the same time say that this occurs mostly in vertical dimension.*

The results section has now been rewritten so hopefully most problems have been solved. The quoted phrase is not inconsistent: all dimensions were elongated, the z dimension more so.

3. *What are the distributions of firing rates of place cells in different recordings apparatuses?*

These are now given in Supp. Data: *Comparison of firing properties between mazes* and Fig. S10 along with other basic analyses comparing characteristics in different mazes.

4. *What is the stability of place cells under the different recording conditions?*

These results can now be found in the Supp. Data: *Recording stability* and Supp. Data: *Field stability*. Corresponding methods can be found in similarly named sections in supplementary

methods. Place cells were stable between the first and second arena. Within sessions cells were generally more stable in the XY plane (in the aligned lattice and arena).

5. *All the analyses related to the 2D arena seem to be unnecessary and might lead to misleading conclusions. For instance, there is a whole paragraph (Lines 162-169) making the point that in a 2D arena there are more place fields along X,Y axis compared to the Z axis. This result is completely inconclusive because there is practically no behavioral coverage in the Z axis in a 2D arena. Isn't it the very reason the authors recorded in a volumetric maze? What is the relevance of the A,B,C axes for the arena? I suggest removing entirely all the analysis related to 2D from the paper. The only reason to include the 2D data would be to compare the statistics of *individual* place cells recorded in both 2D and maze configuration (e.g. is there correlation in place fields size, elongation etc, across different mazes?).*

Our reasoning is that although the arena is a flat horizontal environment the animals were able to move vertically (i.e. through rearing or head movements). Thus, we constructed 3D volumetric firing rate maps for the arena just as we did in the case of the lattice mazes. A clarification for this has been added to Supp. Methods: Field Detection: "Although the open-field arena was flat, animals were free to move vertically in the area (i.e. crouching, standing and rearing) and so we treated the arena data as a thin volume: nevertheless, the analyses for the z dimension (height) must be treated with caution."

Analyses comparing the XY axes of the arena to the Z axis were merely to confirm what readers can already see in the figures and what we would already expect based on the arena being very short as the reviewer points out. This is the same reason for showing results for the ABC analyses in the aligned lattice. We feel that the arena analysis is crucial for comparison to all previously reported 2D experiments in horizontal arenas and gives any reader already familiar with 2D data a foothold on the rest of the data. Otherwise, we have removed repetitive statistics or unnecessary comparisons between non-relevant axes in the results which should hopefully make this text flow a lot more easily.

6. *The analyses related to place-field orientation along different dimensions are potentially interesting, but require much more explanation in the main text, and in the figure legend. It would be useful to show some schematics of how one gets from a traditional firing rate-map to the spheres shown in Fig.2D. This is not a common analysis in the field, and I don't think that the explanations in Line 561 of the Methods are clear enough. For instance – what is the definition of "place-field eigenvectors" (Methods, Line 563?). Also, the description of the behavioral analyses (Methods, 480-488) and the spherical representation in Fig1D are not clear.*

We have added schematics to Fig. 4a and Fig. 7a detailing how these plots are generated. We have also expanded the description given in Supp. Methods: *Field orientation and size* to give more information on the field detection algorithm – this is also accompanied by an explanatory schematic in Fig. S17. The Matlab function we use (*regionprops3*) is a 3D extension of one which is commonly used in place field analysis:

<https://www.ncbi.nlm.nih.gov/pmc/articles/PMC3912569/>.

7. *The results section of the paper often reads like a list of statistical tests, and it is often not clear what should the reader conclude from the different comparisons. Furthermore, many of the relevant analysis are scattered across supplementary figures. It took me almost the entire paper to get to the most important figure – Fig3A. It shows that place-fields are larger in Z (than in X,Y) for aligned but not tilted maze. This result should be highlighted properly, and in my view come before Fig. 2D,E. Analyses of autocorrelation, information content etc take an entire page of the main text while being redundant with the simple point made in Fig3A, and should go to the supplementary.*

Fig. 3A was not intended to show that fields are larger in the Z dimension but instead that fields are less precise in the Z dimension (i.e. they have a larger standard deviation). We do compare the length of fields oriented parallel to the X, Y and Z axes and find that the vertically oriented ones are longest, this is reported much sooner in the results section Results: *Place fields were elongated parallel to the maze axes*. The figure in question is supposed to accompany the autocorrelation, spatial information and binary morphology analyses which independently support the other findings and are an important part of the paper, removing these to supplementary data would risk readers missing an important result.

8. *Line 145 – the results of this section contradicts its caption (“Place fields were elongated in all dimensions”) and suggest that place fields are in fact elongated only in vertical dimension and not in all dimensions (Line 159, FigS5B). So are place fields elongated in other dimension (except vertical?)*

Yes, fields were also elongated parallel to the other maze axes (see the hotspots in Fig. 7B).

9. *In the discussion the authors say again (Line 315): “Elongation did not occur in every direction but was almost always in the direction of the maze/axes boundaries”. I couldn’t find any analyses to back this claim that there is elongation in some other directions, and not in vertical dimension only.*

As above.

10. *Further, elongation in the vertical dimension seem to contradict the claim in the discussion that elongation occurs in the direction of the travel, because vertical dimension is least traveled (in the aligned maze).*

We agree that our explanation for alignment of elongation along maze axes due to their frequency of travel cannot be the whole story, because of the extra elongation in the Z axis.

11. *Fig. S5 B,C – suggest that the fields are elongated along some dimension. Line 155: “With elongated indices that significantly deviated from 1”. Deviations from 1 could result from noisy estimate of the place-field shape, due to firing-rate variability or inhomogeneous behavioral coverage. The author should compare the data to elongation index of simulated spherical fields. This should be done by taking the distribution of place-field firing rates from the data + noise*

(e.g. Poisson noise) and superimposing the spherical fields on actual behavioral trajectories. The authors should also show a histogram of the elongation indices, relative to a control distribution. Same should be done for sphericity index (Fig S5C).

This analysis can now be found in the supplementary methods section 'Field elongation and sphericity' and was adapted from Yartsev and Ulanovsky, (2013) but improved as per reviewer 1's suggestions. Briefly, for each place field we generate spikes according to a multivariate normal distribution inside a perfect sphere with an equivalent diameter to the place field, centred on the centroid of the place field. We then recreate the firing rate map using these data and the animal's original trajectory, and calculate the elongation index and sphericity of the new field. We do this 100 times and compare the observed values to this shuffle.

12. *Can down-sampling of horizontal movements explain the elongation along the Z dimension (Fig. S9)?*

This analysis can now be found in Supp. Methods and Supp. Data: *Trajectory downsampling*. We downsampled trajectory data so that horizontal and vertical movements were matched in frequency. As per reviewer 1's comments we have also extended this to inner/outer movements and top/bottom movements. We do not look directly at place field elongation as this would require generating firing rate maps and calculating place field metrics for every session and downsample type. We instead look at the spatial information and autocorrelation effects reported in text. As neither of these are affected by any shuffle there does not seem to be a reason to expect a difference in place field elongation.

13. *It would be nice to see examples of elongated place fields (not as convex hulls – which are very sensitive to outliers, but as raw data with behavioral trajectory + spikes as well as with firing-rate maps in different projections).*

We have added two figures to supplementary data (Fig. S2 and S3) which show spike plots, firing rate maps and field outlines (not convex hulls) for multiple cells in each lattice maze. These also show projections of the data. Throughout the manuscript we have replaced convex hulls with field outlines where possible, except in Figs. 4&5 where convex hulls are easier to display and discriminate.

14. *I also suggest to make Fig S4, and S5 main figures. In S4, it would be useful to show histograms for data presented in panels B-E, instead of bar-graphs (for the aligned versus tilted maze cases).*

As suggested, we have moved Figs. S4 and S5 to the main text, with some modifications (now main text Figs. 4, 5 & 6). The boxplots have been replaced with joint histogram and point plots (raincloud plots).

15. *From the histogram of field extents in Fig. S5D left – it seems that the distribution of fields extents in Z is bimodal – most of the fields are not elongated (compared to X,Y), but a small proportion are. What is special about these place fields? Do place fields become more elongated*

in Z as a function of their height in the maze? Analysis shown in Fig.S5F should be conducted separately for place-field extent in each dimension versus distance along this dimension.

Because of the weak nature of the bimodality in this distribution we do not feel this effect is strong enough to pursue further. However, it is worth noting that the rightmost peak corresponds approximately to the vertical extent of the lattice and so is likely due to an edge effect of the apparatus clipping fields that we know are longer in the vertical dimension (Results: *Place fields were elongated parallel to the maze axes*).

Field elongation in Z would necessarily have to be related to field centroid in Z. For instance, if a field very close to the bottom of the maze extended vertically into the lattice its centroid (center of mass) would also shift upwards into the maze. The same edge effects can be imagined for the top and sides of the maze also. Overall, this means that elongated fields will have a centroid closer to the maze center due to edge effects rather than a true effect of position. We are not aware of a way to correct for this bias and it is unclear what the hypothesis would be, so we have not included this analysis.

16. *For cells with multiple fields - is there a correlation between size of multiple fields, or their elongation along a specific dimension? Do the authors have the same cells recorded in both tilted and aligned mazes? If so, do cells with elongation in Z in the aligned maze show elongation in the tilted maze?*

The same cells were not recorded in the two mazes and no animal was recorded in both mazes. We have added a section to Supp. Data: *Comparison of firing properties between mazes* and a figure (Fig. S10) showing the relationship between field characteristics in the arena and lattice maze, or between place fields in the lattice mazes. To summarise: fields in the lattice mazes belonging to the same cell do not share similar orientations or sizes at an above chance level; cells with elongated fields in the arena are not more likely to have elongated fields in the lattice mazes; cells with more fields in the arena are not more likely to express more fields in the lattice mazes.

17. *Line 229 . The authors show that spatial and mutual information is higher for the Z dimension, but paradoxically conclude that this means that Z is encoded more poorly. This is a very problematic claim. Larger place-fields of individual cells do not necessarily mean degraded population coding - it depends on factors like the distribution of firing-rates and density of place-fields (see Kim, Ganguli, Frank Journal of Neuroscience 2012).*

Spatial information is higher in slices taken along the Z dimension, and so these are horizontal XY slices (see Fig. S9B, bottom row): we have clarified the axis labels in Fig. 8B to reflect this. If spatial information is higher in these slices it means horizontal spatial coding is higher. The reviewer is correct that larger place fields do not necessarily mean poorer population coding and we have tried to touch on this in the discussion with the multiscale map hypothesis. Our main claim in the manuscript is not that population coding is poorer but that we have observed poorer coding at the single cell level.

18. Fig3E bottom shows that the decoding error is the same for all three axes in all configurations, but the quality factor is different. Why is that? Does the behavioral coverage affect the prior? Would the conclusion be different if the behavior is matched as in Fig S9?

The quality factor does take coverage into account; essentially the decoded positions are compared to decoding composed purely of the average XYZ position coordinate, which will depend on the coverage of the animal. However, due to the small number of sessions we were able to analyse and based on the comments from reviewers we have removed the decoding analysis from the manuscript completely.

19. Line25: "This is important not just for spatial mapping per se but also because the spatial map may form the framework for other types of cognition in which information dimensionality is higher than in real space." Not clear why this is relevant.

We have clarified by amending as follows: "The question arises as to whether this map is three-dimensional, and if so whether the properties are the same in all dimensions, and how information is integrated across these dimensions". We agree that such extrapolation may not be possible, but it remains a valid hypothesis.

20. Line46: Implies a difference between the finding in rats and bats. It is not clear what this difference is, since as the authors mention in a few sentences earlier – Casali et al (ref 7) demonstrated that rat place cells can be isotropic similarly to bat place cells.

This is true – the phrasing was influenced by our findings in grid cells (where a difference from volumetric encoding persists even on the climbing wall). We have rephrased to: "The different patterns of activity in the different types of apparatus could be due to the different movement patterns afforded by the footholds (aligned vs. orthogonal to gravity), or due to the different encoding requirements of travelling on a surface vs. a volume."

21. Fig S1 left, middle should be part of the main figure.
Fig S2 panel F: is left – aligned, right – tilted? Not clear from the legends.

Fig. S1 has been moved to be a part of Fig. 1. Assuming the reviewer means Fig. S3F (as there is no panel F in Fig. S2) the order of these plots has been clarified in the figure legend (this is now Fig. S1g).

22. Fig1, and Fig S2: It's not clear from the schematics what do the rotated axes A,B,C correspond to. Perhaps using different colors/transparent cubes would help.

The axis labels for the schematics in Fig. 2a have been simplified.

23. What is Fig1C? Is it a 2D arena? It's not clear at all that you have a 2D arena and a 3D maze. Showing 3D coverage data for a 2D arena (e.g. "layer" crossing Fig1 C left) is not informative and confusing.

This layer analysis has been removed and the figure has been improved. The leftmost plots do correspond to the arena and now have a column heading to show this.

24. *Fig1A – what is the colorbar in column 1? Looks like blue = many spikes. Red=few spikes. It's confusing given that in column 2 the colorbar seems to be the opposite. What are min/max firing-rates of cells in A column 4?*

Assuming the reviewer means Fig. 2a (there are no spike plots in Fig. 1) this color scheme was inverted by mistake. These spike plots are shown now simply with red spike markers as is more common in the field (i.e. Fig. 3 and Figs. S2 & S3).

25. *Fig. S3 – the two sessions from arena recordings appear as if they correspond to different heights of the maze.*

To improve the example cell figures (and include tilted lattice maze examples as suggested by Reviewer 1) we have removed the arena data for space and for ease of understanding. Aligned data are now shown in Fig. S2, tilted data are shown in Fig. S3.

26. *Line 383 – spatial coding during volumetric navigation in bats does not seem to be conditioned on environmental geometry, whereas the current paper (as well as two previous papers from the same group) shows that in rats the conclusion of any volumetric study strongly depends on the exact geometry of the environment. Therefore, it is unclear how this “undermines” the necessity to study spatial representation in animals such as bats, whose movement in 3D is unconstrained.*

We agree this was poorly worded – we meant to argue against the view that rats don't have a volumetric spatial map and are therefore not suitable for studies of three-dimensional encoding. We have removed the comment about studying bats.

27. *Line 388-387 The logic of this sentence is unclear.*

This section (Methods: *Statistics and figures*) has been reworked and is clearer now.

Reviewer #3:

1. *My major issue with the paper is that it's a bit hard to read. Most of the Results section goes through a wide array of statistics, which all compare three arenas to one another. These are all presented parenthetically in the text in a way that makes the logic hard to follow after a while. My strong suggestion is to make a table of all the statistics that contains three columns for the three environments, and to remove these statistics from the text itself. The text can then be much more concise in summarizing the main results.*

We have greatly cut down the statistics in-text and tried to make this section read more easily. We have moved statistical tests referring to data in the figures to a table (Table 1); the mazes are in rows. Median values and post-hoc test p -values have similarly been removed from text if they can be seen in figures.

2. *Partly related to this, Figure 1 presents data in a somewhat confusing and redundant way. I realize that layer crossings and proportion of total time are different measures, but they are probably strongly correlated to one another. It's not clear why both are presented, and why A,B,C are omitted from Fig 1B but not 1E.*

The layer crossing analysis has been removed completely because the reviewer is correct that it was redundant and confusing. This figure has been generally improved.

3. *The authors keep referring to gravity throughout the paper, and keep switching between referencing gravity and ease of motion. The issue is that the effects of gravity are not demonstrated at all, and the observed effects are likely due to all the movement non-uniformities. There is evidence in the literature for field elongations along common trajectories in other types of environments. But talking about gravity, the authors some sort of a low-level vestibular effect of place cells. I think that it's fine to mention gravity as a possible contribution in the discussion, but that the paper shouldn't be mentioning gravity so casually throughout the text.*

We did not mean by invoking a gravity influence that we imply a vestibular effect, and indeed we don't mention "vestibular" anywhere in the paper. The effect of gravity on spatial coding is evidenced by the difference between the aligned and tilted lattices which differ only in their alignment with respect to the gravity vector (and also by previous findings that horizontal and vertical surfaces are encoded differently). We assume in fact that the mediator of the difference is *not* vestibular, but rather the different locomotor affordances that arise from having struts that are horizontal vs. tilted. We aren't sure in which instances the word is used "casually".

4. *The authors report that autocorrelation values are different between the three slanted axes (lines 194-196). This suggests that a) not enough data was collected to balance our individual rats' idiosyncrasies and/or b) there is some asymmetry in the arena itself that's not accounted for. This asymmetry is not addressed, which is a problem because the result goes against the main logic of the paper (i.e. how can we believe that the Z axis statistics are different due to the vertical alignment of this axis, as opposed to other unknown factors)?*

These tests have been replaced with a more appropriate one; the main problem was the gigantic sample sizes in these figures which result in significant p -values but also very small effect sizes. We now find a difference between aligned XYZ axes but not the tilted ones which is in agreement with our other measures (Fig. S9). It is of course impossible to rule out unknown factors such as trajectory sampling; however, we do not see any indication of this in the behavioural data (Fig. 2), nor when we correct with downsampling (Fig. S6).

5. *In Fig 3E, the quality factor is lower for the Z-axis than for the X or Y axes. This is consistent with the main results of the paper. However, the data come from a total of 4 sessions from a single rat and the effect seems to be driven entirely by a single outlier. It's a bit hard to trust this result. On a related note, are mean or median statistics being used here to compare the axes?*

Due to the small number of sessions we were able to analyse and based on the comments from reviewers we have removed the decoding analysis from the manuscript completely.

Single session figures

Table 1 and main text figures recreated using only 1 session per animal (the session with the most cells). See main text figure legends.

Table 1

Statistical test results

Comparison	Test	Results	Fig.
Proportion of total time along X & Y, arena		$\chi^2(1) = 0.08, p = .78, \eta_p^2 = 0.001$	Fig. 2b
Proportion of total time along X & Y, aligned	FT	$\chi^2(1) = 0.11, p = .74, \eta_p^2 = 0.0041$	Fig. 2b
Proportion of total time along A, B & C, tilted		$\chi^2(2) = 1.50, p = .47, \eta_p^2 = 0.125$	Fig. 2b
Fields per cell, arena, aligned & tilted	KW	$\chi^2(2) = 34.30, p < .0001, \eta_p^2 = .095$	Fig. 4b
Fields per m ³ , arena, aligned & tilted		$\chi^2(2) = 100.05, p < .0001, \eta_p^2 = .277$	Fig. 4c
Field volume, arena, aligned & tilted		$\chi^2(2) = 8.70, p = .0129, \eta_p^2 = .020$	Fig. 4d
Field diameter, arena, aligned & tilted		$\chi^2(2) = 1.8, p = .026, \eta_p^2 = .004$	Fig. 4e
Field elongation, arena, aligned & tilted		$\chi^2(2) = 27.10, p < .0001, \eta_p^2 = .064$	Fig. 6b
Field elongation arena	WSR (compare to 1)	$Z = 11.12, p < .0001, U3 = 0$	Fig. 6b
Field elongation aligned		$Z = 11.20, p < .0001, U3 = 0$	Fig. 6b
Field elongation tilted		$Z = 8.14, p < .0001, U3 = 0$	Fig. 6b
Field sphericity, arena, aligned & tilted	KW	$\chi^2(2) = 100.80, p < .0001, \eta_p^2 = .239$	Fig. 6c
Field sphericity arena	WSR (compare to 1)	$Z = -11.14, p < .0001, U3 = 1$	Fig. 6c
Field sphericity aligned		$Z = -11.20, p < .0001, U3 = 1$	Fig. 6c
Field sphericity tilted		$Z = -8.14, p < .0001, U3 = 1$	Fig. 6c
Field length distributions, aligned	Multiple KS with Bonferroni	X vs Y: $z = 0.12, p > .50$ X vs Z: $z = 0.16, p = .048$ Y vs Z: $z = 0.16, p = .048$	Fig. 6d
Field length distributions, tilted		$p > .99$ in all cases	Fig. 6d
Autocorrelation aligned, X, Y & Z		$\chi^2(2) = 44.10, p < .0001, \eta_p^2 = .105$ X vs Z & Y vs Z, $p < .0001, X$ vs Y, $p > .76$	Fig. 8a
Autocorrelation tilted, A, B & C		$\chi^2(2) = 3.0, p = .225, \eta_p^2 = .018$	Fig. 8a
Proportion of spatial information aligned, X, Y & Z	FT	$\chi^2(2) = 44.30, p < .0001, \eta_p^2 = .10$ X vs Z & Y vs Z, $p < .0001, X$ vs Y, $p > .27$	Fig. 8b
Proportion of spatial information tilted, A, B & C		$\chi^2(2) = 2.8, p = .24, \eta_p^2 = .018$	Fig. 8b
Area under curve, aligned, X, Y & Z		$\chi^2(2) = 7.9, p = .019, \eta_p^2 = 0.015$ X vs Y, $p > .99, X$ vs Z, $p = .022, Y$ vs Z, $p = .11$	Fig. 8c
Area under curve, tilted, X, Y & Z axes and A, B & C axes		$\chi^2(2) = 1.2, p = .56, \eta_p^2 = 0.0044$ $\chi^2(2) = 0.2, p = .92, \eta_p^2 = 0.0006$	Fig. 8c

Test abbreviations and details can be found in Methods: *Statistics*

Figure 2

Figure 4

Figure 5

Figure 6

Figure 7

Figure 8

Figure S8

Reviewers' Comments:

Reviewer #1:

Remarks to the Author:

The authors have substantially overhauled the manuscript, and it is now much improved. However, a few issues still remain.

1. "We looked carefully at the cluster spaces and are confident that cells were not recorded on more than one day, but to be sure, we have also analysed data from only one session per tetrode per rat as suggested. We find the results essentially unchanged. We have added a mention of this to the text:

"Similar results to those reported in main text were also observed when only analyzing one session per animal (the session with the most place cells). These analyses can be replicated using the provided data set and code."

We have also supplied the actual analyses for the reviewers to evaluate (figures available at the end of this document). We felt including these in the paper itself would add unnecessarily to its length."

Cluster spaces being different across days is no guarantee that the neurons recorded on the consecutive days are different. It merely makes ascertaining identity of the neurons harder. The authors must include the figure as a supplementary figure – 20 instead of 19 figures isn't going to increase the length of the supplement substantially. The authors must include the numbers of units recorded in each maze, and the statistics along with this supplementary figure.

2. L152-155 "These shared a similar number of fields (Fig. 7c, overlap with confidence intervals) but in this case the fields aligned with the Z axis were significantly longer (median length, X, Y & Z: 64.2, 57.6 & 78.2 cm, $\chi^2(2) = 26.8$, $p < .0001$, $\eta^2 = .055$, K-W, X vs Y, $p = .074$, X vs Z, $p < .0084$, Y vs Z, $p < .0001$)."

And L158-161 "As before, these orientations were the only ones with more fields than chance (Fig. 7c red shaded area); they shared an equal proportion of fields (Fig. 7c, overlap with confidence intervals) and a similar length (median length, A, B & C: 64.3, 57.9 & 78.2 cm, $\chi^2(2) = 0.80$, $p = .68$, $\eta^2 = .002$, K-W)."

The lengths of XYZ (64.2, 57.6, 78.2 cm) in the first and ABC (64.3, 57.9, 78.2 cm) in the second statement above are suspiciously similar. And yet the stats are completely different. The authors should mention quantiles and clarify to the reader how the same magnitude difference is significant for aligned lattice (significant enough for the authors to make a big deal out of) but not significant for the tilted lattice. Is it just the smaller number of data points in the tilted lattice? Even more worryingly, the lengths of ABC in this version exactly match the lengths of XYZ (64.3, 57.9, 78.2 cm) in the previous version: "However, the fields 176 aligned with these axes differed in length (median length, X, Y & Z: 64.3, 57.9 & 78.2 cm, $\chi^2(2) = 26.8$, $p < .0001$, $\eta^2 = .079$, K-W)"

I suspect that the numbers have been mixed up. I suggest the authors confirm all the numbers through the manuscript to ensure that the correct statistics are reported. It is impossible for the reviewers to ascertain correctness of these stats unless such glaring discrepancies show up.

3. "L183-185 In contrast, there was no significant difference in the tilted lattice when comparing the X, Y and Z axes or A, B and C axes (Fig. 8c, Table 1)."

Neither figure 8 nor table 1 show XYZ comparison for tilted lattice. They only show ABC comparison.

Reviewer #2:

Remarks to the Author:

The authors addressed most of my comments and improved the clarity of the manuscript. I am therefore happy to recommend it for publication.

Reviewer #3:

Remarks to the Author:

The authors have made a serious effort to address my concerns with the previous version. The paper is now much more readable, which was the most serious issue.

One minor comment on the new version: the 2D arena is referred to inconsistently throughout the manuscript. Sometimes the authors refer to it as the "square arena" and sometimes simply "the arena". Because there are three different arenas in the paper, it would help for the terms to be very consistent throughout.

Response to reviewers

Reviewer #1:

1. *Cluster spaces being different across days is no guarantee that the neurons recorded on the consecutive days are different. It merely makes ascertaining identity of the neurons harder. The authors must include the figure as a supplementary figure – 20 instead of 19 figures isn't going to increase the length of the supplement substantially. The authors must include the numbers of units recorded in each maze, and the statistics along with this supplementary figure.*

We have added the figure as requested: we added a section to supplementary data which contains a summary figure and two summary tables which display the main effects observed in the subsampled dataset (Supp. Data: *Subsampled dataset analysis*, Fig. S11, Table S2 and Table S3).

2. *The lengths of XYZ (64.2, 57.6, 78.2 cm) in the first and ABC (64.3, 57.9, 78.2 cm) in in the second statement above are suspiciously similar. And yet the stats are completely different. The authors should mention quantiles and clarify to the reader how the same magnitude difference is significant for aligned lattice (significant enough for the authors to make a big deal out of) but not significant for the tilted lattice. Is it just the smaller number of data points in the tilted lattice? Even more worryingly, the lengths of ABC in this version exactly match the lengths of XYZ (64.3, 57.9, 78.2 cm) in the previous version: "However, the fields 176 aligned with these axes differed in length (median length, X, Y & Z: 64.3, 57.9 & 78.2 cm, $\chi^2(2) = 26.8$, $p < .0001$, $\chi^2(177) = .079$, K-W)" I suspect that the numbers have been mixed up.*

The reviewer is correct; these numbers were mixed up and the lengths provided for the tilted lattice were incorrect. These have been updated to: A, B & C: 69.13, 73.43 & 61.84 cm. The statistical test results were correct and have not been changed.

3. *I suggest the authors confirm all the numbers through the manuscript to ensure that the correct statistics are reported. It is impossible for the reviewers to ascertain correctness of these stats unless such glaring discrepancies show up.*

We have double checked all other statistics reported throughout the manuscript and supplementary data.

4. *Neither figure 8 nor table 1 show XYZ comparison for tilted lattice. They only show ABC comparison.*

This information was provided in Table 1 (last row). For clarity the XYZ and ABC test results have been divided into two rows. We have also added the ABC median values to Table 1 as they are not shown in Fig. 8c.

Reviewer #2:

1. *The authors addressed most of my comments and improved the clarity of the manuscript. I am therefore happy to recommend it for publication.*

We thank the reviewer for their recommendation.

Reviewer #3:

1. *The authors have made a serious effort to address my concerns with the previous version. The paper is now much more readable, which was the most serious issue.*

One minor comment on the new version: the 2D arena is referred to inconsistently throughout the manuscript. Sometimes the authors refer to it as the "square arena" and sometimes simply "the arena". Because there are three different arenas in the paper, it would help for the terms to be very consistent throughout.

We have changed this so it is consistently referred to as the 'arena', except where it is initially described as a "square open field environment ('arena')". We never use arena when referring to the lattice mazes.